# Fine manipulation of sound via lossy metamaterials with independent and arbitrary reflection amplitude and phase

Yifan Zhu [1], Jie Hu[1,2], Xudong Fan[1], Jing Yang[1], Bin Liang[1], Xuefeng Zhu[3] & Jianchun Cheng[1]

The fine manipulation of sound fields is critical in acoustics yet is restricted by the coupled amplitude and phase modulations in existing wave-steering metamaterials. Commonly, unavoidable losses make it difficult to control coupling, thereby limiting device performance. Here we show the possibility of tailoring the loss in metamaterials to realize fine control of sound in three-dimensional (3D) space. Quantitative studies on the parameter dependence of reflection amplitude and phase identify quasi-decoupled points in the structural parameter space, allowing arbitrary amplitude-phase combinations for reflected sound. We further demonstrate the significance of our approach for sound manipulation by producing self-bending beams, multifocal focusing, and a single-plane two-dimensional hologram, as well as a multi-plane 3D hologram with quality better than the previous phase-controlled approach. Our work provides a route for harnessing sound via engineering the loss, enabling promising device applications in acoustics and related fields.

[1] Key Laboratory of Modern Acoustics, MOE, Institute of Acoustics, Department of Physics, Collaborative Innovation Center of Advanced Microstructures, Nanjing University, Nanjing 210093, China. [2] Department of Information Science and Technology, Nanjing Forest University, Nanjing 210037, China. [3] School of Physics, Huazhong University of Science and Technology, Wuhan, Hubei 430074, China. Correspondence and requests for materials should be addressed to B.L. (email: liangbin@nju.edu.cn) or to X.Z. (email: xfzhu@hust.edu.cn) or to J.C. (email: jccheng@nju.edu.cn)

The fine control of three-dimensional (3D) sound field presents a long-standing important question in acoustics, with the fundamental significance to a variety of applications ranging from ultrasound treatment and imaging to architectural acoustics to particle manipulation. The past few years have witnessed considerable efforts dedicated to researching the spatial control of sound, resulting in many methods that enable fascinating wave-steering effects by engineering the phase of transmitted/reflected waves[1–12]. Yet it is still challenging to achieve a complete control of sound that calls for totally independent modulation of all the two degrees of freedom, namely the amplitude and phase of sound, which collectively determine an arbitrary signal. Usually, complex coupling between amplitude and phase has to occur when the incident wave is interacting with sophisticated artificial structures, as evidenced by the fact that previous pure-phase metasurfaces generally ignore the error caused by the unavoidable variation in amplitude as the phase is adjusted[3–11]. Noting that the existing designs are mainly limited to lossless acoustic systems, it is of fundamental interest to explore the possibility to find new physics for sound manipulation by moving into a broader regime with loss of acoustic energy involved[13–20]. Given the conventional perspective that energy loss always destroys wave-steering capabilities and prevents effective downscaling of acoustic devices[13,14], the presence of loss is expected to further complicate such coupling effect.

In this article, we prove, through analytical derivation, numerical simulation, and experimental demonstration, that it is possible to access the decoupled modulation of amplitude and phase of sound by deliberately introducing energy loss in a controlled manner[15–20]. A holey structured lossy acoustic metamaterial (LAM) is proposed to practically implement our design idea based on an inherently different mechanism from the previous ones[3–12], which depend on either passive structures utilizing resonances to only delay the phase of sound or active elements requiring transducer arrays and complicated controlling circuits. The realization of decoupled tuning of amplitude and phase makes all combinations accessible in their full ranges (that is, [0,1] and [0,2$\pi$]), which is necessary for the manipulation of electromagnetic waves as pointed out in previous works on optical metasurfaces[21,22].

By engineering the loss, our proposed mechanism endows the resulting device with the ability to realize the fine control of 3D sound fields, while bearing advantages of simple design, low-cost fabrication, planar profile, high efficiency, and deep-subwavelength resolution. We demonstrate this mechanism by producing some distinct phenomena, such as high-quality Airy beams, multifocal focusing, and both a single-plane two-dimensional (2D) hologram and a multi-plane 3D hologram that are conventionally subject to dramatic deterioration in both the quality and flexibility of the generated sound fields.

## Results

**Decoupled modulation of reflection amplitude and phase.** Figure 1a shows the schematic of LAM comprising an array of unit cells, with predesigned geometries. Here we use absorbing boundaries at the back side to introduce energy loss into the acoustic metamaterial by totally absorbing the controlled leaky wave. The unit cells can modulate both amplitude and phase of reflection at the surface under the illumination of sound on the front side, as indicated by the red arrows in Fig. 1b, where the loss at the back side is required to control the reflection amplitude. As shown in Fig. 1c, each unit cell comprises three components: upper channel ($C_1$), middle channel ($C_2$), and lower channel ($C_3$) with heights of $h_1$, $h_2$, and $h_3$, respectively. The width of channels $C_1$ and $C_3$ is $d = \beta D$ ($\beta$ is defined as the filling ratio of air channel,

$D$ is the period of unit cells) and the width of channel $C_2$ is $w$. The channel walls are assumed to be acoustically rigid. The total height of the unit cell is $h = h_1 + h_2 + h_3$ with $h_2$ fixed as $h_2 = 0.5$ cm. In this study, the operating wavelength is denoted by $\lambda$, the period of unit cells $D = \lambda/4$ and the speed of sound in air $c_0 = 340$ ms$^{-1}$. Therefore, we obtain $D = 0.5$ cm for the chosen operating frequency $f = 17$ kHz. Due to the subwavelength nature of $D$, the amplitude $A$ and phase $\phi$ of reflection are independent of the incidence direction.

In general, $A$ and $\phi$ are related to the structural parameters $w$ and $h_1$ of such a holey metamaterial[22] (Supplementary Notes 1-3, Supplementary Figs. 1-2) and can be expressed as $A = f_1(w, h_1)$ and $\phi = f_2(w, h_1)$, respectively, suggesting the ability to modulate $A$ and $\phi$ by changing both $w$ and $h_1$. However, this is only the first step in accessing all the possible combinations of ($A$, $\phi$). An intuitive way to ensure ergodicity in the phase-amplitude space is to make $A$ and $\phi$ respectively controlled by one parameter, for example, $A = f_3(w)$ and $\phi = f_4(h_1)$, which, as will be revealed later, is enabled by reaching a decoupled point (DP) in the structural parameter space of LAM. In order to give quantitative evaluations of the dependence of reflection amplitude and phase on structural parameters, we define the coupling strengths as

$$M_{A(\phi),h_1(w)} = \frac{\partial A(\phi)}{\partial h_1(w)}. \tag{1}$$

We further obtain the coupling coefficients $\overline{M}_{A(\phi),h_1(w)}$ by integrating the coupling strengths for all combinations of ($h_1$, $w$) and conducting normalization with respect to their maxima (Supplementary Note 4). As aforementioned, a completely decoupled manipulation of reflection amplitude and phase means that the amplitude and phase of reflection should be related to only one structural parameter ($h_1$ or $w$). Here the filling ratio of air channel $\beta$ is a crucial factor to influence $\overline{M}_{A(\phi),h_1(w)}$. For an extreme case of $\beta = 1$, which means the channel walls are infinitely thin, the amplitude $A$ and phase $\phi$ can be analytically derived as (Supplementary Note 2)

$$A = \frac{d^4 - w^4}{d^4 + w^4}, \tag{2}$$

$$\phi = -\frac{4\pi h_1}{\lambda}. \tag{3}$$

Equations (2) and (3) clearly show the existence of completely decoupled relationships characterized by $\overline{M}_{A,h_1} = 0$ and $\overline{M}_{\phi,w} = 0$, respectively, which correspond to DPs for $\overline{M}_{A(\phi),h_1(w)}$. In practice, we can simply access the DPs by bringing $\beta$ close to 1. However, we cannot physically reach them due to the fact that very thin channel walls are flexible and no longer provide a rigid boundary (note that the rigidity of channel walls is the prerequisite condition of all our derivations), and mathematically the whole LAM is transformed into a trivial structure of purely air at $\beta = 1$. When $\beta < 1$, the analytical expressions of $A$ and $\phi$ become quite complicated (Supplementary Note 3). We can still access quasi-decoupling domains, where near-zero coupling coefficients are achievable by setting the parameter $h$ at specific values, which corresponds to quasi-DPs in the structural parameter space. The analytical calculations of $\overline{M}_{A(\phi),h_1(w)}$ vs. structural parameters ($\beta$, $h$) are shown in Fig. 1b. At the DPs where $\beta = 1$, as marked by the cyan crosses, $\overline{M}_{A,h_1}$ and $\overline{M}_{\phi,w}$ are both exactly 0. When $\beta < 1$, the quasi-decoupling condition would occur at some discrete $h$ values, for example, $h = n\lambda/2$ ($n = 2$, 3, 4,...). Apparently, the quasi-decoupling

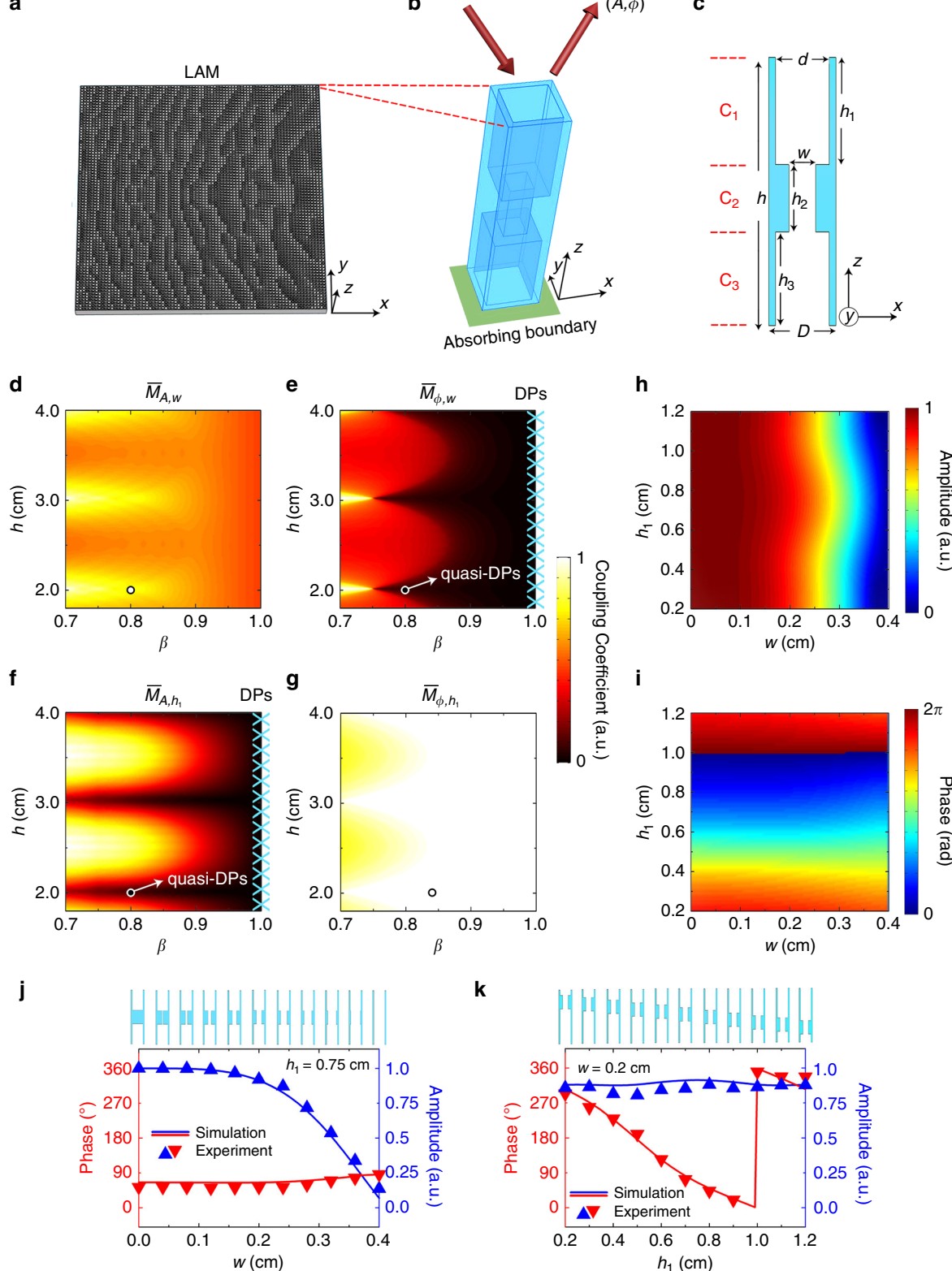

**Fig. 1** Decoupled modulation of reflection amplitude and phase. **a–c** Schematic diagram of holey lossy metamaterials with an absorbing boundary at the back side in **a**, three-dimensional (3D) illustration in **b** and cross-section view of a unit cell in **c** are appended. **d–g** The coupling coefficients $\overline{M}_{A(\phi),h_1(w)}$ vs. $h$ and $\beta$ with decoupled points (DPs) and quasi-DPs marked by the crosses and arrows, respectively. **h, i** The reflection amplitude and phase responses to the parameters $h_1$ and $w$ for a unit cell operating at quasi-DPs. **j, k** The simulated and measured amplitude and phase vs. $w$ and $h_1$, respectively, which reveals that the reflection amplitude and phase are controlled by only one parameter, respectively. The simulations are carried out by the finite element solver in commercial software COMSOL Multiphysics$^{TM}$5.0

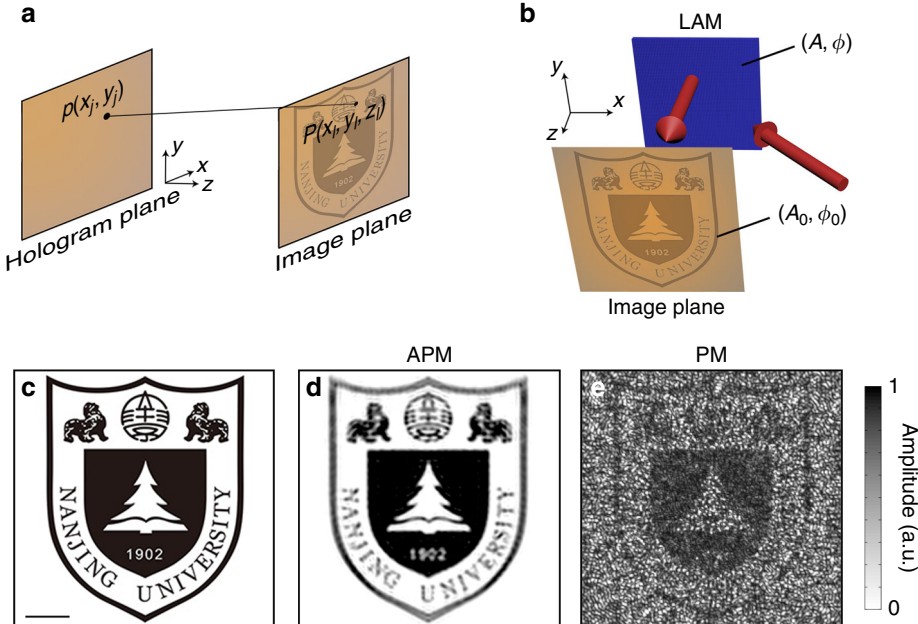

**Fig. 2** High-fidelity acoustic hologram. **a** Schematic diagram of hologram reconstruction. **b** Schematic diagram of how the lossy acoustic metamaterial (LAM) projects high-quality acoustic hologram in simulation and experiment. **c** The target image of Nanjing University logo with complex amplitude distributions. Scale bar, 20 cm. **d** The simulated holographic image by the amplitude-phase-modulation (APM) method. **e** The simulated holographic image by the phase-modulation (PM) method. Permissions for use of the Nanjing University logo in this figure were obtained from Nanjing University. All rights reserved

condition corresponds to the occurrence of Fabry–Pérot resonances.

In consideration of the sample fabrication, we set $\beta = 0.8$ and $h = 2$ cm to achieve the quasi-decoupling condition, as marked by the white circles in Fig. 1d–g. Figure 1h, i presents the correlation of $A$ and $\phi$ to $w$ and $h_1$ at the quasi-DP, that is, $(\beta, h) = (0.8, 2$ cm). Figure 1j, k shows the simulated and measured phase and amplitude responses vs. $w$ and $h_1$ for the two cases of $h_1 = 0.75$ cm and $w = 0.2$ cm, respectively. The results clearly demonstrate the decoupled tuning of $A$ and $\phi$ within the ranges of [0, 1] and [0, $2\pi$]. In the vicinity of quasi-DPs, our designed LAM benefits from the availability of all possible combinations of reflection amplitude and phase by tuning two separate geometric parameters ($h_1$ and $w$) that induce a controlled leaky loss dependent on structural parameters.

It should be pointed out that the energy loss due to thermal viscosity in narrow channels is lower than 1% for each unit cell, since the cross-section of air channels is not deep subwavelength. Therefore, the boundary layer effect can be ignored in the manipulation of reflection amplitude and phase[13,14]. Such a significant feature of independent control of reflection amplitude and phase enables a complete manipulation of sound in 3D space.

**High-fidelity acoustic hologram**. First, we show the fine control of acoustic waves by hologram projection of complicated patterns. In analogy to the optical hologram[23–25], the acoustic hologram projection was only recently reported, which offers new capabilities in particle manipulation and improvement in applications, such as ultrasonic treatment[26–28]. However, due to the lack of capability to modulate both amplitude and phase, the current production of acoustic holograms has to rely on phase-modulation (PM) approaches combined with a complex optimization process[23–29]. Here, we numerically and experimentally show that with decoupled amplitude phase modulation (APM), high-fidelity acoustic hologram can be stably generated without complex optimization process.

Consider the general principle for acoustic hologram reconstruction as illustrated in Fig. 2a. By discretizing the holographic image into a collection of subwavelength image pixels, the acoustic pressure $p_j$ with amplitude $A_j$ and phase $\phi_j$ on the hologram plane (that is, the surface of LAM) can be calculated by superposing the wave components from those image pixels on the image plane, as follows

$$p_j = \sum_{l=1}^{N} \frac{A_{0l}}{r_l} \exp[i(k_0 r_l + \phi_{0l})] \equiv A_j \exp(i\phi_j), \quad (4)$$

where $k_0$ is the wave number, $N$ is the total number of image pixels, $A_{0l}$ and $\phi_{0l}$ are the amplitude and initial phase of the $l$th image pixel at $(x_l, y_l, z_l)$, $r_l = \sqrt{(x_j - x_l)^2 + (y_j - y_l)^2 + z_l^2}$ is the distance between the image pixel and the hologram pixel at $(x_j, y_j, 0)$, $A_j$ and $\phi_j$ are the amplitude and phase of $j$th hologram pixel on the hologram plane, respectively. It is straightforward to project the predesigned acoustic hologram due to time-reversal symmetry. The holographic image can thus be characterized by

$$P(x, y, z) = \sum_{j=1}^{n} \frac{A_j}{r_j} \exp\left[-i(k_0 r_j - \phi_j)\right], \quad (5)$$

where $n$ is the total number of hologram pixels, $r_j = \sqrt{(x - x_j)^2 + (y - y_j)^2 + z^2}$ is the distance between the spatial point at $(x, y, z)$ and the hologram pixel at $(x_j, y_j, 0)$ in the hologram plane. However, the previous schemes for producing acoustic holograms assume $A_j = 1$ and the holographic image is then characterized by

$$P'(x, y, z) = \sum_{j=1}^{n} \frac{1}{r_j} \exp\left[-i(k_0 r_j - \phi'_j)\right]. \quad (6)$$

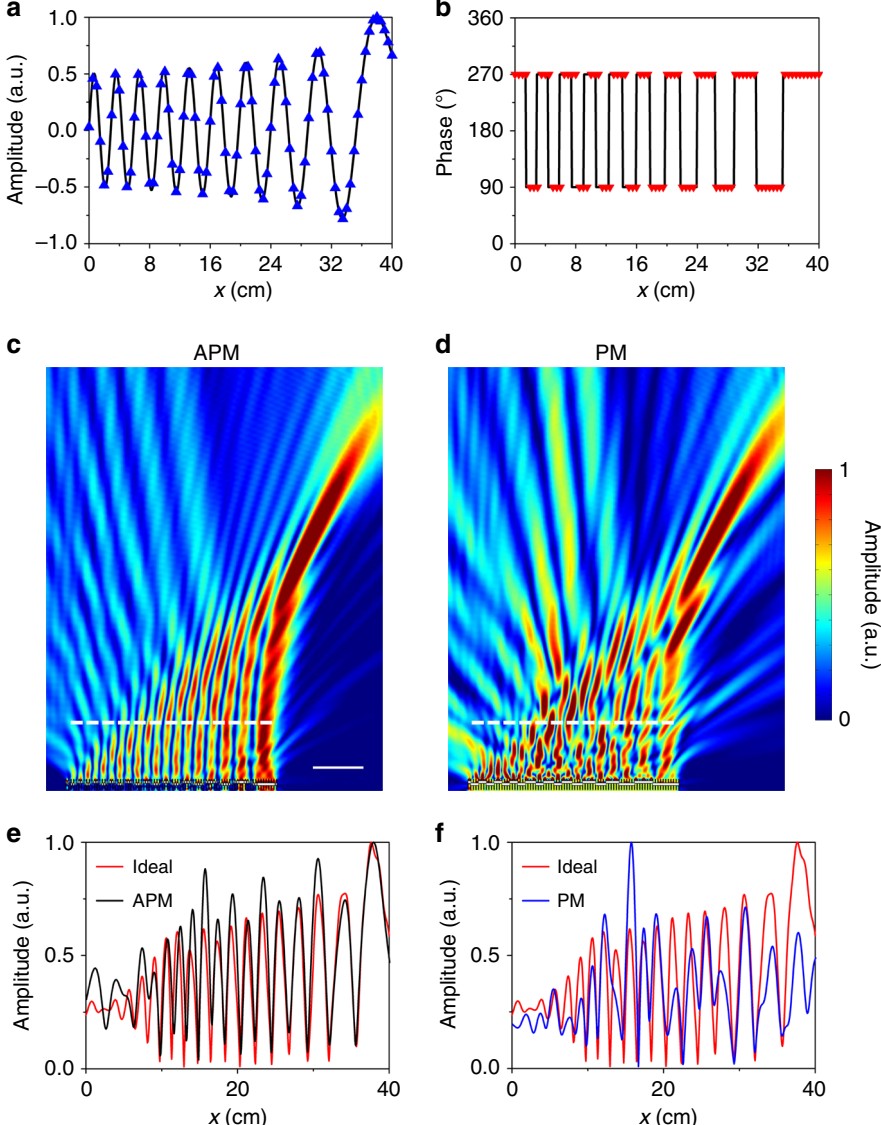

**Fig. 3** Formation of high-quality Airy beam. **a**, **b** Amplitude profile in **a** and phase profile in **b** at the surface of the LAM for generating the Airy beam. The LAM consists of 80 discrete unit cells (triangles). **c**, **d** Simulated pressure fields of Airy beams produced by the APM in **c** and PM in **d**. Scale bar, 10 cm. **e**, **f** Distributions of pressure amplitude on the dashed lines in **c** and **d**. The result for the perfect Airy beam is calculated by Eq. (7), which is displayed for comparison (red lines)

Consequently an optimizing algorithm is needed to generate the target phase $\phi'_j$ for active phased arrays[29] or pure-modulation masks[26,27]. However, we point out that even for a very simple target image, the optimized hologram amplitude $A_j$ is an inhomogeneous distribution ranging from 0 to 1. Thus, it is important to introduce independent modulations of amplitude and phase, which is straightforward by controlling $(A_j, \phi_j)$ at the hologram plane to generate $(A_{0l}, \phi_{0l})$ at the image plane. Here, the reflection direction is normal to the hologram plane, as shown in Fig. 2b. Our method allows for the production of arbitrarily complex holographic images, such as those with continuously changing gray values, which may be impossible for previous methods that only engineer the phase of sound.

In order to show the superiority of independent APM, we have chosen a complicated predesign image for numerical investigation. The target image is the pattern of a school badge in Fig. 2c. We record the phase and amplitude information independently into the hologram plane for projecting the image. In Fig. 2d, the numerical result at the image plane shows that the holographic

image is of high-quality compared with the target one. For previous PM methods[26–28], the complexity of phase optimization and the quality of holograms are determined by the fineness of target images. Here, we append the holographic image simulated by the PM optimization of Gerchberg–Saxton algorithm in Fig. 2e. Comparing Fig. 2d, e, our APM method outperforms the PM method; in light of time-reversal symmetry it is necessary to modulate both the reflection amplitude and phase for achieving an exact hologram reconstruction of a complex image. We also note that in Fig. 2e, the image error caused by a uniform amplitude distribution on the hologram plane becomes quite prominent when the target image is complicated and comprises a large number of pixels with uneven amplitude levels.

The result demonstrates the effectiveness and flexibility of our method in complicated hologram reconstruction. Simulations for a more complicated hologram are provided in Supplementary Fig. 3 to further reveal the advantage of the APM method. In simulations, the number of pixels in the hologram plane for projecting the hologram in Fig. 2d is 359 × 359, which is a very

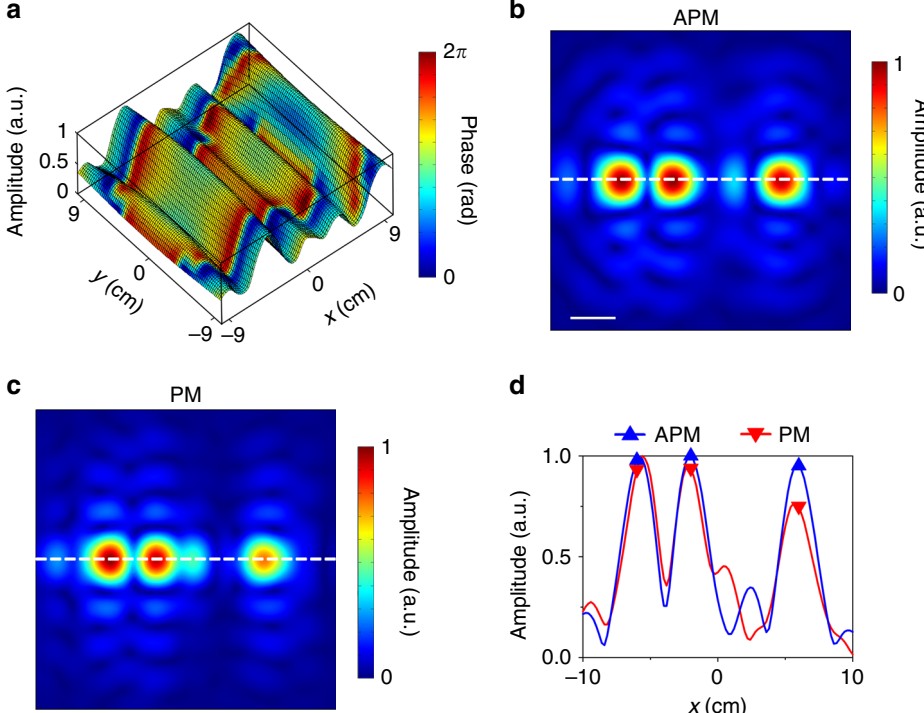

**Fig. 4** Multifocal acoustic focusing. **a** Amplitude and phase profiles at the surface of the LAM for multifocal acoustic focusing. **b** Simulated multifocal acoustic focusing by the APM method. Scale bar, 3 cm. **c** Simulated multifocal acoustic focusing by the PM method. **d** Distributions of pressure amplitude on the dashed lines in **b** and **c**. Three target focal points are marked by triangles

large value. Yet the success of our proposed scheme in generating high-fidelity acoustic holograms is by no means a limit of its potential application. The engineering of loss is indeed significant for controlling sound fields at will and may have applications in varied scenarios. As typical examples, we will further show the production of the Airy beam, multifocal focusing, and both a single-plane 2D hologram and a multi-plane 3D hologram via holey structured LAMs in the following.

**High-quality Airy beam and multifocal focusing**. Next, we employ a LAM to generate an Airy beam based on the method of APM, in comparison with the one formed by the PM method[30,31]. Figure 3a, b shows the reflection amplitude and phase profiles on the surface of the LAM (along the $x$ direction) required for generating a high-quality Airy beam. The spatial distribution of pressure amplitude follows an Airy function as shown in Fig. 3a, with the corresponding phase profile displayed in Fig. 3b[30]. Figure 3c, d provide the comparison of the projected sound-field distributions between the APM and PM methods, where it is apparent to the naked eyes that the Airy beam formed by APM has a better quality than the one formed by PM.

To quantitatively evaluate the quality of the generated beam, we also plot in Fig. 3e, f the distribution of pressure amplitude along the dashed lines in Fig. 3c, d, that is $5\lambda$ away from the sample surface. The result shows a good agreement between the Airy beam generated by APM and the ideal one calculated by[32]

$$p = \sum_{j=1}^{N} \text{Ai}\left(\frac{x_j}{a}\right) H_0^{(1)}(k_0 r_j), \quad (7)$$

where $\text{Ai}(x_j/a)$ is the Airy function, $a$ is a constant, $H_0^{(1)}$ is the zero-order Hankel function of the first kind. In comparison, a much larger deviation of the Airy beam generated by PM from the ideal one is observed in the right part of Fig. 3e, f. These

results suggest that our design with the ability to yield decoupled manipulation of reflection amplitude and phase has the potential to improve the quality of non-diffraction beams as compared with the previous pure-phase method, such as using metasurfaces.

We have also quantitatively analyzed how APM helps to improve the quality of multifocal acoustic focusing in comparison with PM, where the size and number of unit cells of the two samples are $19.6 \times 19.6 \times 2 \text{ cm}^3$ and $39 \times 39$, respectively. Both metamaterial samples are designed to generate a uniform multifocal focusing in an image plane parallel to and 20 cm away from the sample surface. The three focal points locate at $(-6, 0 \text{ cm})$, $(-2, 0 \text{ cm})$, and $(6, 0 \text{ cm})$, respectively. Figure 4a shows the reflection amplitude and phase profiles at the surface of the LAM with the APM approach, which is calculated based on Eq. (4). The simulated acoustic pressure fields for APM and PM are shown in Fig. 4b, c, respectively. Figure 4d shows the pressure amplitude distribution curves along the dashed lines penetrating all three preset foci, as denoted in Fig. 4b, c, where the triangles mark the values at the three target focal points. Our numerical results clearly reveal that the APM method can produce highly uniform sound intensity at the three focal points, agreeing well with the design. Comparatively, although the pure-phase method is good at mimicking a simple single-focus effect, the intensities of different foci generated by the PM method are subject to substantial fluctuations for the case of multiple focusing. These results evidence the significance of APM for the production of complex sound field that actually needs a precise control of not only the phase but also the pressure amplitude on the hologram plane.

**Experiments of single-plane and multi-plane holograms**. In this section, we choose a tree image (Fig. 5a) as our target object, comprising $200 \times 200$ image pixels. Figure 5b presents the reflection amplitude and phase profiles on the hologram plane for projecting the tree pattern in the far field. The calculations of amplitude and phase profiles are based on Eq. (4). In the

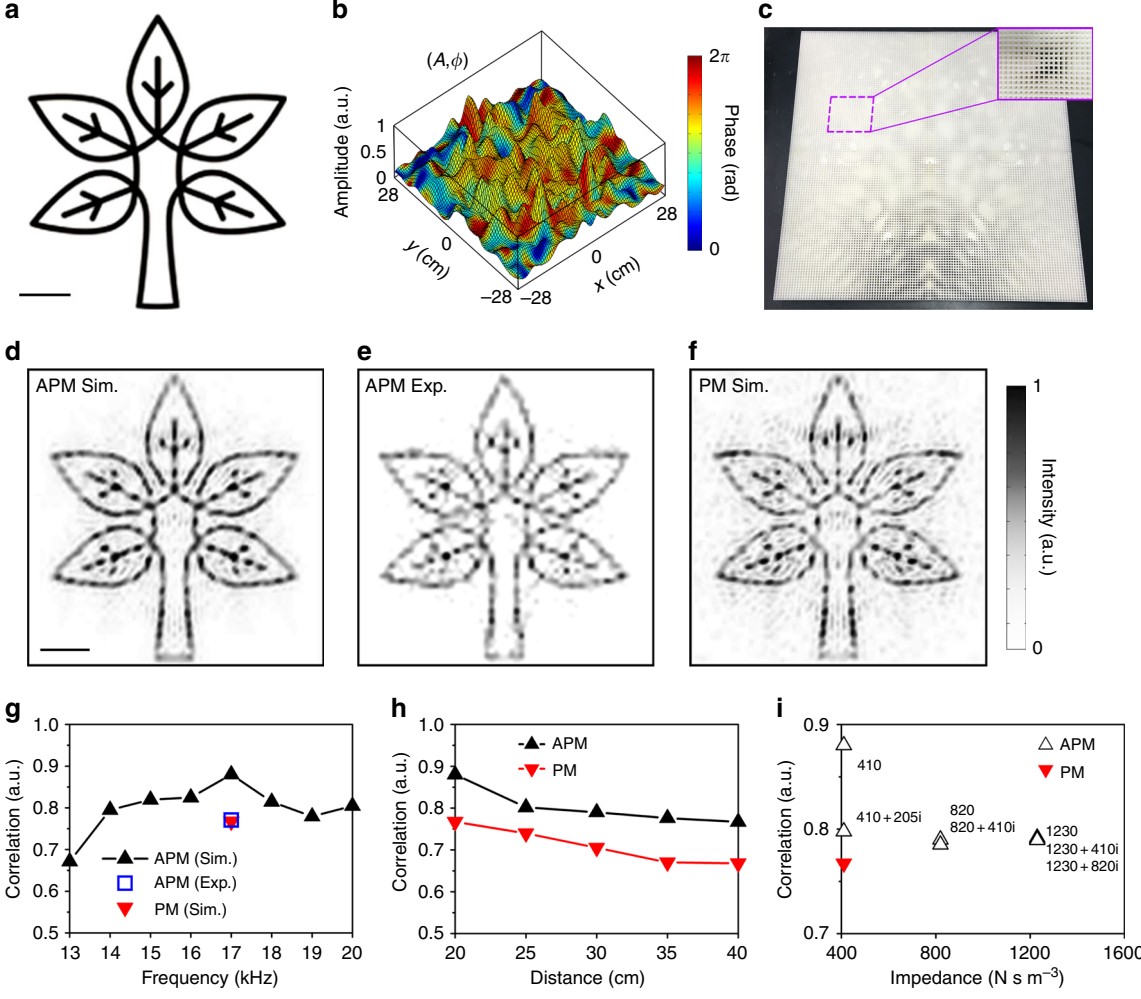

**Fig. 5** Experimental verification of single-plane acoustic hologram. **a** The predesigned image of a tree. Scale bar, 10 cm. Rights for use of this image are from Dreamstime (www.dreamstime.com). All rights reserved. **b** Amplitude and phase profiles on the hologram plane for projecting the tree image. **c** The photograph of the 3D printed LAM sample. **d, e** The simulated holographic image by the APM method and the experimentally measured result. Scale bar, 10 cm. **f** The simulated holographic image by the PM method. **g** The correlation between the resulting image and the predesigned image at different frequencies from 13 to 20 kHz for APM, and at 17 kHz for PM. **h** The correlation between the resulting image and the predesigned image when the image plane locates at different distances. **i** The correlation between the resulting image and the predesigned image for different back impedances

experiment, we fabricated the LAM samples via 3D printing with precision of 0.1 mm. The experiments were carried out in an anechoic chamber to demonstrate the acoustic hologram projection. We record both amplitude and phase information into the LAM sample, where the sample size is $60 \times 60 \times 2$ cm$^3$ with $119 \times 119$ unit cells, as shown by the photo in Fig. 5c. The size of image area is $60 \times 60$ cm$^2$, with a distance 20 cm away from the surface of LAM. Other experimental details can be found in the Methods section. Due to the size limitations in 3D printing, the pixel number of the target image in our experiment is less than the numerical investigations in Fig. 2.

We plot the simulated and measured intensity distributions on the image plane in Fig. 5d, e, respectively, showing a good agreement between numerical and experimental results of fine single-plane 2D hologram. Figure 5f shows the simulated result based on the PM method for comparison. For a quantitative evaluation of the quality of acoustic hologram, we introduce the parameter image correlation which has been commonly used for measuring the similarity between the numerical/experimental image and the target one[33]. The calculation of correlation can be referred to the Supplementary Note 5. A higher value of correlation

denotes a better similarity between the generated holographic image and the target image, and only when the two images are completely identical can a unitary correlation be achieved.

For the simple pattern shown in Fig. 5a, the PM method leads to less obvious errors (as shown in Fig. 5f), since the amplitude distribution via the APM method on the hologram plane is relatively uniform (as shown in Fig. 5b). The improvement by APM (in Fig. 5d) is still evident by both the naked-eye visual effect and the quantitative evaluation of image correlation. Figure 5g shows the relation between image correlation and operation frequency. The results reveal that our designed LAM has a relatively broad operation bandwidth, with the best performance observed at 17 kHz, due to the low dispersion of the groove structure[32]. Since the quasi-DP locates at 17 kHz, the image correlation in simulation (experiment) reaches maximum of 0.880 (0.771). We also note that the holographic image based on APM in a broad frequency range (14–20 kHz, correlation >0.770) is better than the one of the PM method (17 kHz, correlation = 0.767). The simulated holographic images at different frequencies based on PM or APM are appended in the Supplementary Fig. 4.

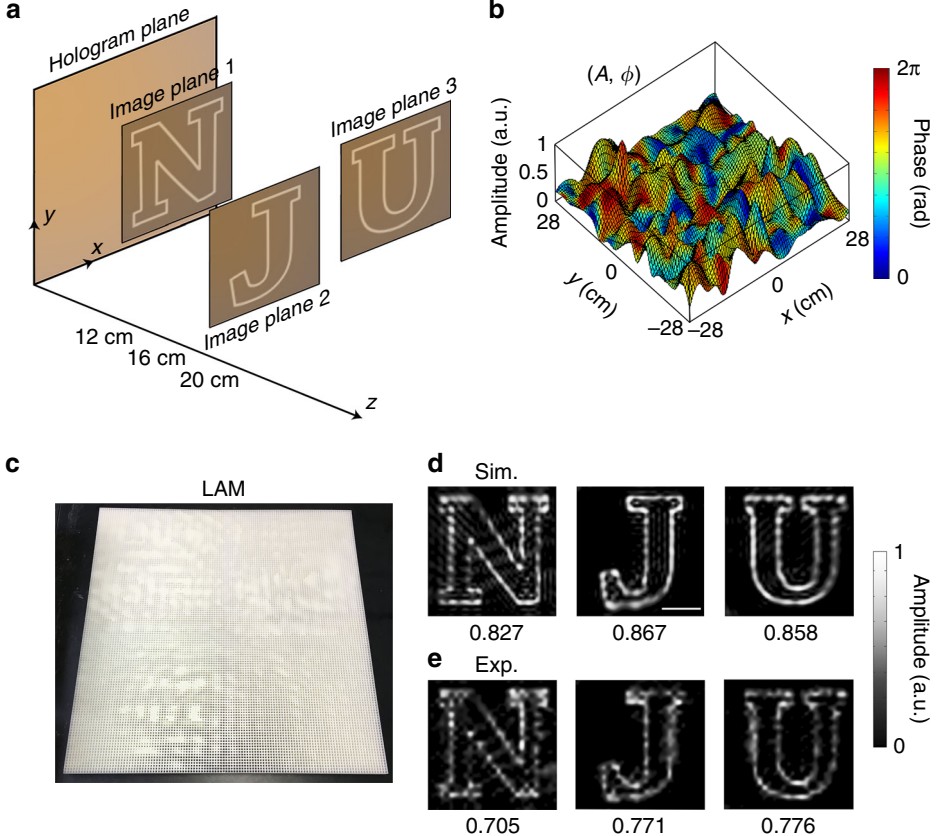

**Fig. 6** Experimental verification of multi-plane acoustic hologram. **a** The predesigned image of a multi-plane acoustic hologram (letters "N," "J," and "U" at different distances of 12, 16, and 20 cm). **b** Amplitude and phase profiles on the hologram plane for projecting the "N," "J," "U" images at multiple planes. **c** The photograph of the 3D printed LAM sample. **d**, **e** The simulated holographic images by the APM method and the corresponding experimentally measured results. The correlations are marked in the figure. The correlation between the resulting image and the predesigned image is appended below each sub-figure. Scale bar, 10 cm. Rights for use of the NJU font images in this figure are from Adobe Font Software License: https://www.fontspring.com/lic/ztxtzqqpzi. 

Figure 5h plots the image correlation at different distances of holographic image planes. The effect for APM is better than that of PM, and the correlation slightly decreases with larger distances due to wave diffraction. Even though the small size of unit cells ($\sim 0.25\lambda$) helps to generate continuous amplitude and phase profiles, the method does not allow breaking the resolution limit in the hologram plane. However, Fig. 5h points out that we can still achieve high fidelity when the image plane is set close to the sample surface. Moreover, we emphasize simultaneous control of reflection amplitude and phase can be achieved even when the back absorption is partial (Supplementary Fig. 5). In this case, we can also project holograms with relatively good correlations to the target image, as unveiled in Fig. 5i.

Last, we demonstrate an acoustic hologram at multiple planes, as schematically depicted at Fig. 6a, where the holographic images are designed to be letters "N," "J," and "U" at three different planes that are spacing 12, 16, and 20 cm away from the hologram plane. The size of holographic regions at image planes 1, 2, and 3 is $60 \times 60$ cm$^2$, and the bottom left corners of those holographic regions locate at (0, 30 cm), (10, 0 cm) and (30, 20 cm) in the x–y plane. We record amplitude and phase distributions (Fig. 6b) into the LAM sample of $119 \times 119$ unit cells (Fig. 6c). By comparing the amplitude field patterns in simulations and experiments in Fig. 6d, e, we observe a good agreement. To be specific, the image correlations to the perfect cases of letters "N," "J," and "U" are 0.827(0.705), 0.867(0.771), and 0.858(0.776) for the results of simulations (experiments), respectively.

## Discussion

In summary, we have shown that by judiciously controlling the leaky loss in acoustic metamaterials, one can achieve the fine control of acoustic waves, by modulating both amplitude and phase of acoustic waves in a static, precise, and decoupled manner. To exhibit the powerful ability of LAM in wave manipulation, we report the numerical and experimental realization of high-fidelity acoustic holograms and demonstrate the production of arbitrarily complicated acoustic fields based on the new mechanism for achieving decoupled amplitude and phase distributions. We also show that our method significantly improves the quality of other kinds of spatial manipulation of sound fields over previous pure-phase methods. We envision that our findings on tailoring loss for harnessing sound fields will have various acoustic-wave-based applications. For example, the realization of complicated 3D holographic images[23] can produce arbitrarily shaped focusing or spatial field patterns for the high-intensity focused ultrasound treatments[34] that hitherto have to rely on the time-consuming scanning method, which is also intriguing for particle manipulation[26,29], architectural acoustics[32], and stereo sound-field reconstruction[35], etc.

## Methods

**Sample fabrication and experiment measurement**. All the samples are made of photosensitive resin, and are manufactured via 3D printing technique (SLA, 0.1 mm in precision). The experiment for measuring reflection amplitude and phase responses of different unit cells is performed in a commercial impedance tube

system (Brüel&Kjær Impedance Measurement Tube for ASTM Type 4206A, Brüel&Kjær PULSE Multi-analyzer system Type 3560, and Brüel&Kjær Power Amplifier Type 2716, Supplementary Note 6, Supplementary Fig. 6). The experiments for acoustic holograms in Figs. 5e and 6d are performed in an anechoic chamber for reducing undesired sound reflection. In order to generate a plane wave, an 8-cm-radius loudspeaker driven by a multifunctional signal generator (RIGOL DG1022) is located in the far field, about 3 m away from the center of LAM sample. The measured region is set at the image plane with an area of $60 \times 60$ cm$^2$. The leaky back of the sample is facing towards the sound-absorbing panels that are set 2 cm away from the sample, the same as the simulation case. The scanning of pressure amplitude fields is conducted by using a 1/4-inch-diameter Brüel&Kjær type-4961 microphone and Brüel&Kjær PULSE Type 3160.

**Numerical simulation**. 3D numerical simulations are carried out by the finite element solver in commercial software COMSOL Multiphysics$^{TM}$5.0, using a high performance computing cluster. A full 3D geometry is used in the "Pressure Acoustics, Frequency Domain" module. Perfectly matched layers are imposed on the outer boundaries of simulation domains to prevent reflections[36].

**Data availability**. The data that support the findings of this study are available from the corresponding author upon request.

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

## Acknowledgements

The authors appreciate fruitful discussions with Prof. Cheng-wei Qiu and Prof. Jensen Li. This work was supported by the National Key R&D Program of China, (grant no. 2017YFA0303700), National Natural Science Foundation of China (grant nos. 11634006, 81127901, 11674119, 11404125, and 11690030). Y.Z. is supported by the program A for Outstanding PhD candidate of Nanjing University.

## Author contributions

Y.Z., J.H., X.F., and J.Y. performed the theoretical simulations; Y.Z., J.H., and X.F. designed and carried out the experiments; Y.Z., X.Z., B.L., and J.C. wrote the manuscript; X.Z., B.L., and J.C. guided the research. All authors contributed to data analysis and discussions.

## Additional information

**Competing interests:** The authors declare no competing interests.

