## [Peer Review File · Nature Communications]

Reviewers' comments:

Reviewer #1 (Remarks to the Author):

The authors in this work have designed and realized an acoustic metasurface for reflection holograms by employing an inhomogeneous profile of resonance tubes.

By tuning both 2 geometric parameters together, one on resonance condition (sliding the neck of the structure) to control the reflection phase and another on coupling (width of neck) to a loss channel to control the amplitude, they obtain total degree of freedom of both amplitudes (zero to one) and phases (full 2π range). One can think the addition of loss can control the amplitude but the smart point of the design is that the loss component is not controlled by addition of different amount of loss materials but simply by the coupling to a background loss. Moreover, amplitude and phase can be controlled independently by exploring a concept of decoupling point, which can be achieved by choosing the right size and filling fraction of the whole structure. This is a very nice and generic concept to construct holograms. Surely this is done in acoustics in this work and I can see its generality and simplicity of the underlying principle and design may be useful for other domains as well.

When I read up to Fig. 2 to see the effect of the hyperfine control of acoustic images due to the control of amplitudes, I am quite impressed but it seems that the experimental results in Fig. 5 does not quite match the expectation of complexity of the object to be displayed. Is it only a limitation due to the number of pixels employed in this work? If the authors can further improve on this, like a 3D hologram or an image of finer resolution, that will keep the excitement and make a large difference to previous approaches.

When it comes to the experimental results, I guess you are using the same frequency (17kHz?) in the numerical simulations. However, it is worth to say it clearly to have a rough idea on the different parameters, like the size of unit cell, 2 cm wavelength, 20 wavelength away for the image, etc. There are two aspects that the authors should discuss. One on frequency dispersion. Will the design work with a reasonable bandwidth? Another is on the error analysis. A more quantitative analysis, e.g. rms error, should be done on comparing simulation and experimental results.

For the simulations, how is the hologram and airy beam simulated? with or without the structural unit cell? That should be clarified.

Please also clarify in text how the absorbing boundary is realized in experiment. Is it just an open boundary or some absorbing materials there?

On a whole, I found the manuscript very enjoyable to read, seeing its potential on applications and suitable for broad readership. I would really like to recommend its publication after the authors improve on the above issues.

Reviewer #2 (Remarks to the Author):

This manuscript describes a means to model acoustic metamaterials (AMM) that permit the independent control of both the magnitude and phase of a reflected signal using simple structured elements. The authors provide a detailed description of the behavior of these elements and the approach to determining the specific geometries that enable arbitrary control of magnitude and phase of the acoustic signal reflected from their surface. The authors state that the primary contribution of

their work is the consideration of loss in the elements and demonstration of the ability to provide full control of the reflected field despite this loss. This is indeed a unique contribution in terms of the existing acoustic metamaterial research and is therefore of interest for publication in Nature Communications. However, it is the opinion of this reviewer that several points and ambiguities need to be addressed prior to acceptance. Those points are listed below.

1) The manuscript considers losses at the back of the metamaterial elements (as described in lines 87-89 of page 5). This approach does indeed take into account the loss in the elements, but it seems too simplistic for the claims that are made in the manuscript. Specifically, the authors claim that their model clearly shows that when losses are present, regardless of losses in the AMM structures. However, the losses considered here are only for the case of a perfectly absorbing boundary at the back of the elements. It is not at all clear what this means for more general losses. The following cases should be discussed and probably analyzed in a revised manuscript if the authors wish to keep the strong statement that this work is in regards to "lossy metamaterials" in general.

a. What happens if the impedance at the back of the AMM structures is not perfectly absorbing, but instead consists of some complex impedance, $Z_{\text{back}} = Z_r + jZ_i$? Can the model consider this case and still achieve the arbitrary control? This must be clearly addressed in the revision. It would be best if results from one case be shown in comparison with the current results.

b. More importantly, unless I have missed something, the present manuscript only considers loss at the back boundary, and not losses induced within the elements. Such losses, thermos-viscous in nature, are distributed within the AMM structures. It is not clear that the AMM structure, the design scheme, and the modeling is sufficient to capture these types of losses and whether they are important are not. The authors need to clearly address this point as it is highly relevant to their central points.

2) The term "leaky loss" is used throughout the manuscript. What, precisely, is meant by this term in the context of this particular case? Do the authors mean that the AMM leaks energy out the back of the hologram plane? More details need to be provided or a different term should be employed.

3) One of the key claims that the authors make is that the independent control of magnitude and phase allows for improved control of the pressure fields. This indeed seems to be the case. However, the authors do not provide any discussion on the resolution limitations of their approach in terms of wavelength. What is the minimum size of the structures at the hologram plane? Does this approach simply allow us to have a higher fidelity control (as evidenced by their results), but not to surpass standard resolution limits? A discussion on these points needs to be provided in the revised manuscript.

4) Figures 2c and 2d would be more compelling if they included the image reconstruction for both amplitude and phase control AMM and just phase controlled AMM. The current figure is good, but it lacks an ability to provide a qualitative comparison between the two different approaches.

5) Finally, it is not clear from this work why including loss at the back of the structure is necessary to get independent control of amplitude and phase. Is this truly necessary? Can it be done without losses being present? Please provide a discussion of this in the revision.

Minor points to address

1) The first sentence in the abstract should be re-written. It's seems a bit too grandiose for a scientific publication

2) Similarly, the use of the term "hyperfine" in the title seems a bit too strong of a statement. It would seem that the term 'fine' would be better.

3) Why did the authors define the coupling strengths in Eq. (1) in terms of both geometric variables rather than defining four strengths like $M_{\{A,h\}} = (\partial A)/(\partial h)$, $M_{\{A,w\}} = (\partial A)/(\partial w)$, ...? As they are currently defined, the coupling strength can be zero if it has no dependence on either variable, but gives no information on the dependence of h and w independently?

The current definition seems to work for the design, but it seems to hide information. It would be best if the authors could provide a comment on this point in the manuscript when those parameters are introduced.

4) Line 126 of page 7 has a discussion about the case where $\beta = 1$ and the fact that it cannot be hit in reality because of the finite impedance contrast between air and elastic solids. Isn't the $\beta = 1$ case where portions of the AMM structure is purely air? The impedance contrast doesn't seem relevant.

5) Aren't the patterns shown in Fig 1b Fabry-Perot types of resonances? Please address in the revision.

Reviewer #3 (Remarks to the Author):

The paper "Hyperfine manipulation of sound via lossy acoustic metamaterials" by Zhu, Hu, Fan, Yang, Liang, Zhu, and Cheng reports the manipulation of both the amplitude and phase across the wavefront of an acoustic wave incident on a planar metamaterial made of discrete sub-wavelength elements. The key feature reported is the introduction of a loss (amplitude change) via controlled leaky emission from the backside of each element. The authors successfully determine the requirements for independently setting the complex amplitude and phase at each element, which leads to the demonstration of holograms that can now encode both amplitude and phase. This is in principal an interesting piece of work and an advance in the field of acoustics, as it suggests improvements in the generation of sound fields. However, these improvements are mainly shown in simulations and do not manifest themselves in the actual experiments. Important information is missing and the claimed universal improvements are not demonstrated. Therefore further work is needed and the authors are asked to address the following points:

1) Title: „hyperfine“ has a special meaning in physics. How does it relate to this work? The authors probably mean high fidelity. However, the title should be changed. Independent control of the static amplitude and phase across an acoustic wavefront is the essence of this work and this should be reflected in the title.

2) The approach the authors present is limited to reflection. The scalability, especially miniaturization, is limited by two factors, (a) the fabrication method and (b) the requirement of full absorption (or the disappearance) of transmitted wave components at the backside. Considering these limitations the results are not "universal" and are not as spectacular as the authors claim. The text should be changed accordingly.

3) The work mainly shows via simulations that the control of amplitude and phase improves holograms. This is well known from optics. The paper does not appear to demonstrate any (real) improvement in the experimental acoustic fields. A convincing experimental demonstration is missing and should be provided by the authors so that the importance of the work can be judged.

4) Please, add scale bars or coordinate axes. This applies to almost all images and plots.

5) How are the phase-only results (PM) obtained, against which the APM are compared? Do you use an optimization procedure or simply keep the phase of the APM and reset all amplitudes to 1? How does this compare to optimized PM of other published works? This information must be provided.

6) It is not clear what "Freewheeling" means (abstract).

7) p.6, Equation 1: capital M is used for both coupling strengths and transfer matrices in the SI. This

is an unnecessary source of confusion and the nomenclature should be changed.

8) p.6, L.116: What does $(M_A)^-(M_\Phi)^-=0$ mean? Is it $(M_A)^-=(M_\Phi)^-=0$?

9) p.7, L.138: Do you mean Supplementary Note 3 or 4? Regarding Supp. Note 4, why do you integrate w over $[0, 0.4]$ and h over $[0.2, 1.2]$? One would expect the ranges $[0, \beta D]$ and $[0, \lambda/2]$, respectively.

10) On p.8 the authors write that "However, due to the lack of capability to modulate both amplitude and phase, the current production of acoustic holograms ...cannot guarantee high-fidelity of images". This does not seem to be correct as phase-only holograms have been shown to generate extremely high-fidelity images?

11) p. 10, L.192ff: Please choose a number of unit cells that allows comparison to either your experimental data or previously published hologram data. The images in Figure 2 are phenomenal but so is the element count of 359×359 . The experimental data presented in Figure 5 look mediocre compared to what has been achieved with pure phase holograms in other works.

12) p.12, L.245: Reference to equation 3 not 5.

13) p.13, L.265: The Penrose pattern is shown in Figure S3.

14) At various locations throughout the manuscript and in the conclusions the authors speak of "modulating both amplitude and phase of acoustic wave in a precise, continuous and decoupled manner". This is somewhat misleading as continuous modulation suggests a temporal or dynamic control. The authors should clarify this by stating clearly in the text that they only consider fixed or static acoustic holograms.

**Referee #1 (Remarks to the Author):**

1. *The authors in this work have designed and realized an acoustic metasurface for*
*reflection holograms by employing an inhomogeneous profile of resonance tubes.*

*By tuning both 2 geometric parameters together, one on resonance condition (sliding*
*the neck of the structure) to control the reflection phase and another on coupling (width*
*of neck) to a loss channel to control the amplitude, they obtain total degree of freedom*
*of both amplitudes (zero to one) and phases (full 2π range). One can think the addition*
*of loss can control the amplitude but the smart point of the design is that the loss*
*component is not controlled by addition of different amount of loss materials but simply*
*by the coupling to a background loss. Moreover, amplitude and phase can be controlled*
*independently by exploring a concept of decoupling point, which can be achieved by*
*choosing the right size and filling fraction of the whole structure. This is a very nice*
*and generic concept to construct holograms. Surely this is done in acoustics in this*
*work and I can see its generality and simplicity of the underlying principle and*
*design may be useful for other domains as well.*

*When I read up to Fig. 2 to see the effect of the hyperfine control of acoustic images*
*due to the control of amplitudes, I am quite impressed but it seems that the experimental*
*results in Fig. 5 does not quite match the expectation of complexity of the object to be*
*displayed. Is it only a limitation due to the number of pixels employed in this work? If*
*the authors can further improve on this, like a 3D hologram or an image of finer*
*resolution, that will keep the excitement and make a large difference to previous*
*approaches.*

**Response:** We thank the referee for the positive remarks and valuable advices. In light
of the referee's report, we have made every effort to revise and improve the manuscript.
It is true that holographic images can be improved by increasing the number of pixels.
Following the suggestion of the referee, we have further increased the number of pixels
insofar as the size of samples does not exceed the limit of our 3D printing machine, and
added the experimental demonstration of projection of a 2D image with finer resolution

as well as production of fine distribution of acoustic energy in 3D space. A quantitative
comparison between the images generated by the proposed amplitude-phase
modulation (APM) method and by the previous phase modulation (PM) method has
also been made to clearly show the merits of our scheme. Specifically, we have
fabricated new lossy acoustic metamaterials (LAM) samples consisting of 119×119
unit cells and conducted experiments on projecting a single-plane 2-D hologram with
finer resolution and multi-plane 3-D hologram as shown respectively in Figs. 5 and 6
in the updated version of manuscript. We have also added discussions for clarification.

Please refer to

**Pages 13-15, lines 265-326:**

**Experimental verification of single-plane 2-D hologram and multi-plane 3-D**
**hologram.** In this section, we choose a tree image [Fig. 5(a)] as our target object,
comprising 200×200 image pixels. Figure 5(b) presents the reflection amplitude and
phase profiles on the hologram plane for projecting the tree pattern in the far field. The
calculations of amplitude and phase profiles are based on Eq. (3). In the experiment, we
fabricated LAM samples via 3-D printing with precision of 0.1mm. The experiments
were carried out in an anechoic chamber to demonstrate the acoustic hologram
projection. We record both amplitude and phase information into the LAM sample,
where the sample size is $60 \times 60 \times 2 \text{cm}^3$ with 119×119 unit cells, as shown by the photo
in Fig. 5(c). The size of image area is $60 \times 60 \text{cm}^3$, with a distance 20cm away from the
surface of LAM. Other experimental details can be found in the Methods part. Due to
the size limitations in 3-D printing, the pixel number of the target image in our
experiment is less than the numerical investigations in Fig. 2.

We plot the simulated and measured intensity distributions on the image plane in
Figs. 5(d) and 5(e), respectively, showing a good agreement between numerical and
experimental results of fine 2-D hologram. Figure 5(f) shows the simulated result based
on the PM method for comparison. For a quantitative evaluation of the quality of
acoustic hologram, we introduce the parameter of “image correlation” which has been
commonly used for measuring the similarity between the numerical/experimental
image and the target one. The calculation of correlation can be referred to the

Supplementary Note 5. A higher value of correlation denotes a better similarity between
the generated holographic image and the target image, and only when the two images
are completely identical can a unitary correlation be achieved, which represents a
perfect hologram. Figure 5(g) shows the relation between image correlation and the
operation frequency. The results reveal that although our LAM is designed to work at
17kHz, it has a relatively broad operation bandwidth, thanks to the low dispersion of
the groove structure [33]. At 17kHz where the quasi-decoupled point (quasi-DP) locates,
the image correlation reaches a maximum of 0.880 in simulation, and the corresponding
measured data, albeit much lower than the simulated one due to the unavoidable
experimental error, still reaches 0.771 and is higher than the ideal value one can achieve
with PM method. We also note that the holographic image based on APM in a broad
frequency range (14kHz~20kHz, correlation>0.770) is better than the one of the PM
method (17kHz, correlation=0.767). The simulated holographic images at different
frequencies based on PM or APM are appended in the Supplementary Fig. 4. Notice
that our proposed method does not surpass standard resolution limits, which is in theory
the only limitation on its performance of sound manipulation. Hence the size of each
unit cell at the hologram plane is chosen as 1/4 wavelength, which is sufficiently small
for avoiding spatial alias and generating smooth phase and amplitude profiles. This
important feature, together with the independent control of magnitude and phase,
enables controlling acoustic waves with a higher fidelity control, especially when the
image plane is not far away from the sample. Figure 5(h) illustrates the comparison
between the image correlations as functions of the distances of holographic image
planes for APM and PM methods. Clearly, the hologram quality for APM is always
much better than that of PM regardless of the distance of image plane, although for both
cases the correlation slightly decreases with larger distances due to wave diffraction, as
shown in Fig. 5(h). Moreover, we emphasize simultaneous control of reflection
amplitude and phase can be achieved even when the back absorption is partial (see
Supplementary Fig. 5). In this case, we can also project holograms with relatively
higher correlations to the target image, as unveiled in Fig. 5(i).

At last, we demonstrated both numerically and experimentally the production of

precise distribution of acoustic energy in 3-D space. Here we choose to project the
acoustic hologram onto multiple planes instead of a single 2-D plane, as schematically
depicted at Fig. 6(a), where the holographic image is designed to be three hollow letters
“N”, “J”, and “U” at three different planes that are spacing 12cm, 16cm, 20cm away
from the hologram plane. The size of holographic regions at image planes 1, 2 and 3 is
$60 \times 60 \text{cm}^3$, and the bottom left corners of those holographic regions locate at (0cm,
30cm), (10cm, 0cm) and (30cm, 20cm) in the x - y plane. We record amplitude and phase
distributions [Fig. 6(b)] into the LAM sample of 119×119 unit cells [Fig. 6(c)]. By
comparing the amplitude field patterns in simulations and experiments, we
unambiguously observe a very good agreement. To be specific, the image correlations
to the perfect cases of letters “N”, “J”, “U” are 0.827(0.705), 0.867(0.771), 0.858(0.776)
for the results of simulations(experiments), respectively.

[Editorial Note: Image redacted from Peer Review File to avoid copyright infringement.]

Figure 5 | Experimental verification of single-plane 2-D acoustic hologram. (a) The
predesigned image of a tree. (b) Amplitude and phase profiles on the hologram plane
for projecting the tree image. (c) The photograph of the 3-D printed LAM sample. (d-
e) The simulated holographic image by the APM method and the experimentally

measured result. (f) The simulated holographic image by the PM method. (g) The
correlation between the resulting image and the predesigned image at different
frequencies from 13kHz to 20kHz for APM method, and at 17kHz for PM method. (h)
The correlation between the resulting image and the predesigned image when the image
plane locates at different distances. (i) The correlation between the resulting image and
the predesigned image for different back impedances.

[Editorial Note: This image has been redacted to avoid copyright infringement.]

Figure 6 | Experimental verification of multi-plane 3-D acoustic hologram. (a) The predesigned image of a multi-plane acoustic hologram (Letters “N”, “J”, “U” at different distances of 12cm, 16cm, 20cm). (b) Amplitude and phase profiles on the hologram plane for projecting the “N”, “J”, “U” images at multiple planes. (c) The photograph of the 3-D printed LAM sample. (d-e) The simulated holographic images by the APM method and the corresponding experimentally measured results. The correlations are marked in the figure. The correlation between the resulting image and the predesigned image is appended below each sub-figure.

2. When it comes to the experimental results, I guess you are using the same frequency
 (17kHz?) in the numerical simulations. However, it is worth to say it clearly to have a
 rough idea on the different parameters, like the size of unit cell, 2 cm wavelength, 20
 wavelength away for the image, etc. There are two aspects that the authors should
 discuss. One on frequency dispersion. Will the design work with a reasonable
 bandwidth? Another is on the error analysis. A more quantitative analysis, e.g. rms
 error, should be done on comparing simulation and experimental results.

**Response:** In the revised version, we define a parameter of “image correlation” on the
 error analysis for quantitatively evaluating the quality of acoustic hologram. The image
 correlation has been commonly used to measure the degree of similarity between the
 numerical/experimental image and the target one, where the mathematical definition
 can be referred to the Supplementary Note 5. Please refer to

**“Note 5. Calculation of the correlation between two images.**

The correlation for evaluating the similarity between two images is calculated by

$$140 \text{ Correlation} = \frac{\sum_m \sum_n (A_{mn} - \bar{A})(B_{mn} - \bar{B})}{\sqrt{\left(\sum_m \sum_n (A_{mn} - \bar{A})^2\right)\left(\sum_m \sum_n (B_{mn} - \bar{B})^2\right)}}, \quad (\text{S28})$$

where A and B are the data matrices of the two images, and \bar{A} and \bar{B} are the mean
 values of the elements in the matrices A and B , respectively.” on pages 10-11, lines
 136-140 in the supplementary materials. Based on the definition, a unitary correlation
 denotes that the two images are identical, and the holographic image is perfect.

By utilizing the parameter of “image correlation”, we quantitatively investigate the
 bandwidth of our design work as well as the robustness of performance against
 distances of holographic image planes. In the revised manuscript, Fig. 5(g) shows the
 relation between “image correlation” and the operation frequency. The results reveal
 that our designed LAM has a relatively broad operation bandwidth, with the best effect
 observed at 17kHz. Since the quasi-DP locates at 17kHz, the image correlation in
 simulation (experiment) reaches maximum of 0.880(0.771). We also note that the

holographic image based on APM in a broad frequency range (14kHz~20kHz,
correlation>0.770) is better than the one of the PM method (17kHz, correlation=0.767).
The simulated holographic images at different frequencies based on PM or APM are
appended in the Fig. S4 of supplementary materials. Figure 5(h) plots the image
correlation at different distances of holographic image planes. Clearly, the effect for
APM is better than that of PM, and the correlation slightly decreases with larger
distances due to wave diffraction.

Figure 5(g). The correlation between the resulting image and the predesigned image at
different frequencies from 13kHz to 20kHz for APM, and at 17kHz for PM.

[Editorial Note: This image has been redacted to avoid copyright infringement.]

**Figure S4 | The holographic images calculated by the PM or APM method at different frequencies. PM: Phase modulation. APM: Amplitude and phase modulation.**

Figure 5(h). The correlation between the resulting image and the predesigned image
when the image plane locates at different distances.

3. For the simulations, how is the hologram and airy beam simulated? with or without
the structural unit cell? That should be clarified.

**Response:** Thank you for pointing out this issue. The high-fidelity hologram in Fig. 2
is simulated by the effective parameters (amplitude and phase), since the required
number of pixels (359×359) on the hologram plane is huge and our computing cluster
cannot support the full-wavelength simulation of such a large model. Other results in
Figs. 3-6 (the Airy beam, multi-focal focusing, single-plane 2-D hologram as well as
multi-plane 3-D hologram) are simulated with the modeled LAM comprising structural
unit cells. Please refer to Fig. 2 with a revised caption and the remark “As typical
examples, we will further show the production of the Airy beam, multi-focal focusing,
sing-plane 2-D hologram as well as multi-plane 3-D hologram via holey structured
LAM in the following.” on **page 11, lines 220-222** in the revised manuscript.

[Editorial Note: This image has been redacted to avoid copyright infringement.]

Figure 2 | High-fidelity acoustic hologram. (a) Schematic diagram of hologram reconstruction. (b) Schematic diagram of how LAM projects high-quality acoustic hologram in simulation and experiment. (c) The target image of a school badge with

complex amplitude distributions. (d) The simulated holographic image by the APM
method. (e) The simulated holographic image by the PM method. (f-h) Another case of
projecting a more complicated acoustic hologram with the target image an Einstein's
photo. The simulation is conducted by effective parameters.

4. Please also clarify in text how the absorbing boundary is realized in experiment. Is
it just an open boundary or some absorbing materials there?

**Response:** Thank you for pointing out this issue. In the revised manuscript, we point
out that the leaky back of our sample is facing towards the sound-absorbing panels that
are set 2cm away from the sample, the same as the case in the full-wave simulation.
Please refer to “The leaky back of the sample is facing towards the sound-absorbing
panels that are set 2cm away from the sample, the same as the simulation case.” on
**page 17, lines 357-358** in the revised manuscript.

5. On a whole, I found the manuscript very enjoyable to read, seeing its potential on
applications and suitable for broad readership. I would really like to recommend its
publication after the authors improve on the above issues.

**Response:** Thank you for your appreciation on our work.

**Referee #2 (Remarks to the Author):**

1. *This manuscript describes a means to model acoustic metamaterials (AMM) that*
*permit the independent control of both the magnitude and phase of a reflected signal*
*using simple structured elements. The authors provide a detailed description of the*
*behavior of these elements and the approach to determining the specific geometries that*
*enable arbitrary control of magnitude and phase of the acoustic signal reflected from*
*their surface. The authors state that the primary contribution of their work is the*
*consideration of loss in the elements and demonstration of the ability to provide full*
*control of the reflected field despite this loss. This is indeed a unique contribution in*
*terms of the existing acoustic metamaterial research and is therefore of interest for*
*publication in Nature Communications. However, it is the opinion of this reviewer that*
*several points and ambiguities need to be addressed prior to acceptance. Those points*
*are listed below.*

**Response:** We thank the referee for the positive remarks and valuable advices. We have
made every effort to revise and improve the manuscript.

1) *The manuscript considers losses at the back of the metamaterial elements (as*
*described in lines 87-89 of page 5). This approach does indeed take into account the*
*loss in the elements, but it seems too simplistic for the claims that are made in the*
*manuscript. Specifically, the authors claim that their model clearly shows that when*
*losses are present, regardless of losses in the AMM structures. However, the losses*
*considered here are only for the case of a perfectly absorbing boundary at the back of*
*the elements. It is not at all clear what this means for more general losses. The following*
*cases should be discussed and probably analyzed in a revised manuscript if the authors*
*wish to keep the strong statement that this work is in regards to “lossy metamaterials”*
*in general.*

a. *What happens if the impedance at the back of the AMM structures is not perfectly*
*absorbing, but instead consists of some complex impedance, $Z_{\text{back}} = Z_r + jZ_i$?*

*Can the model consider this case and still achieve the arbitrary control? This must be*
*clearly addressed in the revision. It would be best if results from one case be shown in*
*comparison with the current results.*

**Response:** Thank you for those important questions and suggestions. To answer the
referee’s question, we first define a parameter of “image correlation” on the error
analysis for evaluating the quality of acoustic hologram. The “image correlation”
measures the degree of similarity between the numerical/experimental image and the
target one, where the mathematical definition can be referred to the Supplementary
Note 5. Please refer to

**“Note 5. Calculation of the correlation between two images.**

The correlation for evaluating the similarity between two images is calculated by

$$250 \text{Correlation} = \frac{\sum_m \sum_n (A_{mn} - \bar{A})(B_{mn} - \bar{B})}{\sqrt{\left(\sum_m \sum_n (A_{mn} - \bar{A})^2\right)\left(\sum_m \sum_n (B_{mn} - \bar{B})^2\right)}}, \quad (\text{S28})$$

where A and B are the data matrices of the two images, and \bar{A} and \bar{B} are the mean
values of the elements in the matrices A and B , respectively.” on pages 10-11, lines
136-140 in the supplementary materials. Based on the definition, a unitary correlation
denotes that the two images are identical, and the holographic image is perfect.

By utilizing the parameter of “image correlation”, we quantitatively investigate the
performance of our approach when the back impedance is complex or not perfectly
absorbing. The correlations at different back impedances are shown in Fig. 5(i) in the
revised manuscript.

**[Editorial Note: This image has been redacted to avoid copyright infringement.]**
**Figure 5 | Experimental verification of single-plane 2-D acoustic hologram.** (a) The
pre-designed image of a tree. (b) Amplitude and phase profiles on the hologram plane
for projecting the tree image. (c) The photograph of the 3-D printed LAM sample. (d-
e) The simulated holographic image by the APM method and the experimentally
measured result. (f) The simulated holographic image by the PM method. (g) The
correlation between the resulting image and the pre-designed image at different
frequencies from 13kHz to 20kHz for APM method, and at 17kHz for PM method. (h)
The correlation between the resulting image and the pre-designed image when the image
plane locates at different distances. (i) The correlation between the resulting image and
the pre-designed image for different back impedances.

Note that the simulated holographic images in Figs. 5(d) and 5(f) are corresponding
to the cases at the hollow triangle with back impedance $410\text{N}\cdot\text{S}/\text{m}^3$ (correlation=0.880)
and at the red triangle (correlation=0.767) in Fig. 5(i), respectively. The results in Fig.
5(i) show that we can still achieve very good holographic images (correlation>0.767)
when the impedance at the back of the LAM structure is not perfectly absorbing. In this

case, we can conduct simultaneous (but may not be independent) control of amplitude
 and phase, as unveiled in the added Fig. S5 of supplementary materials. A totally
 independent control of reflection amplitude and phase is achieved at specific back
 impedance as predicted by our theoretical analysis as well as by the numerical results
 corresponding to $Z=410 \text{ N}\cdot\text{S}/\text{m}^3$. This simply means a perfect matching of impedance
 and can be conveniently realized in practice by just keeping the back of each unit cell
 open (if there is relatively a large space behind the sample) or by placing a perfect
 absorptive panel near the backside of sample (which is the very way we used in
 experiment and is necessary when the sample needs to be attached to a rigid wall).

 **Figure S5 | The calculated reflection amplitude and phase for different back**
 **impedances. (a) $Z=410 \text{ N}\cdot\text{S}/\text{m}^3$, (b) $Z=820+410i \text{ N}\cdot\text{S}/\text{m}^3$, (c) $Z=1230+410i \text{ N}\cdot\text{S}/\text{m}^3$.**

b. *More importantly, unless I have missed something, the present manuscript only*
*considers loss at the back boundary, and not losses induced within the elements. Such*
*losses, thermos-viscous in nature, are distributed within the AMM structures. It is not*
*clear that the AMM structure, the design scheme, and the modeling is sufficient to*
*capture these types of losses and whether they are important are not. The authors need*
*to clearly address this point as it is highly relevant to their central points.*

**Response:** Thank you for this very important point. Following the suggestion of the
referee, we have conducted simulations by incorporating the thermal viscosity within
each unit cell into account and find out that the energy loss due to thermos-viscous
effect is lower than 1%. Our result agrees with the acoustic theory, since the thinnest
channels in our designed structure are still orders of magnitude larger than the thickness
of boundary layer despite the subwavelength scale of the whole unit cell. As a result,
the thermal-viscous effect in LAM structures is trivial and will not appreciably affect
the manipulation of amplitude and phase, which is also verified by the good agreement
between the simulations and measurements. In the revised manuscript, we have added
some discussions on this issue. Please refer to “It should be pointed out that the energy
loss due to thermal viscosity in narrow channels is lower than 1% in numerical
simulations, since the cross section of air channels is still much larger than the thickness
of boundary layer. Therefore, the thermal-viscous effect in LAM structures is trivial
and will not appreciably affect the independent manipulation of amplitude and phase,
which is also verified by the good agreement between the simulations and
measurements.” on pages 7-8, lines 152-157.

2) *The term “leaky loss” is used throughout the manuscript. What, precisely, is meant*
*by this term in the context of this particular case? Do the authors mean that the AMM*
*leaks energy out the back of the hologram plane? More details need to be provided or*
*a different term should be employed.*

**Response:** In our work, the term “leaky loss” refers specifically to the energy leaking

out the back of the hologram plane, which will not be reflected back due to the matched
impedance that can be realized conveniently in practice by simply leaving the back of
each unit cell open (if there is relatively large space behind the sample) or by placing a
perfect absorptive panel near to the backside of our sample (which is the way we used
in experiment and is useful when the sample needs to be attached to a rigid wall). In the
revised manuscript, we have provided more details on this issue. Please refer to “The
leaky back of the sample is facing towards the sound-absorbing panels that are set 2cm
away from the sample, the same as the simulation case.” on **page 17, lines 357-358.**

*3) One of the key claims that the authors make is that the independent control of*
*magnitude and phase allows for improved control of the pressure fields. This indeed*
*seems to be the case. However, the authors do not provide any discussion on the*
*resolution limitations of their approach in terms of wavelength. What is the minimum*
*size of the structures at the hologram plane? Does this approach simply allow us to*
*have a higher fidelity control (as evidenced by their results), but not to surpass standard*
*resolution limits? A discussion on these points needs to be provided in the revised*
*manuscript.*

**Response:** As the referee points out, our method does not surpass standard resolution
limits, which is in theory the only limitation on its performance of sound manipulation.
Hence for our sample, the size of each unit cell at the hologram plane is chosen as $1/4$
wavelength, which is sufficiently small for avoiding spatial alias and generating smooth
phase and amplitude profiles. This important feature, together with the independent
control of magnitude and phase, enables controlling acoustic waves with a higher
fidelity control, especially when the image plane is not far away from the sample. For
clarification we plot in Fig. 5(h) in the revised manuscript the comparison between the
image correlations as functions of the distances of holographic image planes for APM
and PM. Clearly, the hologram quality for APM is always much better than that of PM
regardless of the distance of image plane, although for both cases the correlation slightly
decreases with larger distances due to wave diffraction, as shown in Fig. 5(h).

Figure 5(h). The correlation between the resulting image and the predesigned image
 when the image plane locates at different distances.

We have also added some discussions on this issue. Please refer to “Figure 5(h)
 illustrates the comparison between the image correlations as functions of the distances
 of holographic image planes for APM and PM methods. Clearly, the hologram quality
 for APM is always much better than that of PM regardless of the distance of image
 plane, although for both cases the correlation slightly decreases with larger distances
 due to wave diffraction, as shown in Fig. 5(h). Moreover, we emphasize simultaneous
 control of reflection amplitude and phase can be achieved even when the back
 absorption is partial (see Supplementary Fig. 5). In this case, we can also project
 holograms with relatively higher correlations to the target image, as unveiled in Fig.
 5(i).” on page 14, lines 305-313.

4) *Figures 2c and 2d would be more compelling if they included the image
 reconstruction for both amplitude and phase control AMM and just phase controlled
 AMM. The current figure is good, but it lacks an ability to provide a qualitative
 comparison between the two different approaches.*

**Response:** In light of the referee’s suggestion, we add the hologram results based on
 optimized phase modulation (PM) in Figs. 2(e) and 2(h) for comparison. In addition to
 the quantitative comparison displayed in the new Fig. 5, we have added in the updated

version for numerically and experimentally showing the merits enabled by independent
control of phase and amplitude, the results in Fig. 2 provide a qualitative comparison
between the two different approaches and clearly give a visual demonstration of how
the proposed APM method outperforms the PM method. It is apparent that the
generation of holographic images with great complexity is really challenging for
optimized PM yet can be achieved with high fidelity by our APM, which however, is
difficult to realize experimentally within a limited time.

[Editorial Note: This image has been redacted to avoid copyright infringement.]

Figure 2 | High-fidelity acoustic hologram. (a) Schematic diagram of hologram reconstruction. (b) Schematic diagram of how LAM projects high-quality acoustic hologram in simulation and experiment. (c) The target image of a school badge with complex amplitude distributions. (d) The simulated holographic image by the APM method. (e) The simulated holographic image by the PM method. (f-h) Another case of projecting a more complicated acoustic hologram with the target image an Einstein's photo. The simulation is conducted by effective parameters.

In addition, we add some discussions on a qualitative comparison between the two
different approaches in the revised manuscript. Please refer to “Here, we append the
holographic image simulated by PM optimization of Gerchberg-Saxton (GS) algorithm
in Fig. 2(e). Comparing Figs. 2(d) and 2(e), our method clearly outperforms the PM
method, providing a great flexibility in hologram reconstruction. The second target
image is an Einstein's photo with different gray values, where the amplitudes at image
pixels are continuously varied between 0 and 1, as shown in Fig. 2(f). The holographic
image in Fig. 2(g) based on APM is consistent with the target image, while the
holographic image based on PM is very blurred, as shown in Fig. 2(h).” on **pages 10,**
**lines 207-214** in the revised manuscript.

*5) Finally, it is not clear from this work why including loss at the back of the structure*
*is necessary to get independent control of amplitude and phase. Is this truly necessary?*
*Can it be done without losses being present? Please provide a discussion of this in the*
*revision.*

**Response:** When manipulating the reflected acoustic waves, the reflection amplitude
would always be unity in the absence of energy loss. The presence of loss effect is
therefore necessary for the production of non-unitary magnitude but does not guarantee
better performance of acoustic manipulation due to the ubiquitous coupling between
the amplitude and phase variation. The essence of our current work lies in that we have
proved both theoretically and experimentally that leaking loss effect, if engineered
properly by using specific geometries, could lead to independent and arbitrary control
of amplitude and phase, enabling high-fidelity manipulation of acoustic waves. In the
revised version, following the suggestion of the referee, we have added new results for
investigating how the quality of acoustic manipulation by the proposed APM depends
on the back impedance and provided some discussions on this issue. The added results
quantitatively prove that the introduction of loss effect in our proposed metastructure
enables simultaneous control over the amplitude and phase and helps to improve its

wavefront-steering capability, while a totally-independent amplitude and phase control
for producing the best effect needs to be achieved when the leaky loss at the back is
perfect. In the revised manuscript, we have provided some discussions on this issue.
Please refer to “The unit cells are capable to modulate both amplitude and phase of
reflection at the surface under the illumination of sound on the front side, as indicated
by the red arrows in Fig. 1(a), where the loss at the back side is required to get control
of reflection amplitude. Here we would like to mention that the reflection amplitude
would always be unitary in the absence of energy loss when manipulating the reflected
acoustic waves. The presence of loss effect is therefore necessary for the production of
non-unitary magnitude but does not guarantee better performance of acoustic
manipulation due to the ubiquitous coupling between the amplitude and phase variation.
The essence of this work lies in that the leaking loss effect, if engineered properly by
using specific geometries, could lead to independent and arbitrary control of reflection
amplitude and phase, enabling high-fidelity manipulation of acoustic waves.” on **pages**
**4-5, lines 86-97** in the revised manuscript.

Minor points to address:

1) *The first sentence in the abstract should be re-written. It's seems a bit too grandiose*
*for a scientific publication*

**Response:** We have rewritten the first sentence in the abstract. Please refer to “**Fine**
**manipulation of sound field in 3-D space is an important issue in acoustics but hitherto**
**is restricted by the coupled amplitude and phase modulations in existing wave-steering**
**metamaterials.**” on **page 2, lines 24-26** in the revised manuscript.

2) *Similarly, the use of the term “hyperfine” in the title seems a bit too strong of a*
*statement. It would seem that the term ‘fine’ would be better.*

**Response:** We have made a careful check throughout the manuscript and changed the
term “**hyperfine**” into “**fine**” throughout the manuscript based on the suggestion.

3) Why did the authors define the coupling strengths in Eq. (1) in terms of both
geometric variables rather than defining four strengths like $M_{\{A,h\}} = (\partial A / \partial h)$,
$M_{\{A,w\}} = (\partial A / \partial w)$, ...? As they are currently defined,
the coupling strength can be zero if it has no dependence on either variable, but gives
no information on the dependence of h and w independently? The current definition
seems to work for the design, but it seems to hide information. It would be best if the
authors could provide a comment on this point in the manuscript when those parameters
are introduced.

**Response:** In the revised manuscript, we have defined four coupling strengths in Eq.
(1) in light of the referee's suggestion, as follows

$$M_{A,h_1} = \frac{\partial A}{\partial h_1}, M_{A,w} = \frac{\partial A}{\partial w}, M_{\phi,h_1} = \frac{\partial \phi}{\partial h_1}, M_{\phi,w} = \frac{\partial \phi}{\partial w}, \quad (1)$$

We further obtain the coupling coefficients $\overline{M}_{A(\phi),h_1(w)}$ by integrating the coupling
strengths for all combinations of (h_1, w) and conducting normalization with respect to
their maxima (See Note 4 in the revised supplementary materials). As aforementioned,
a completely decoupled manipulation of reflection amplitude and phase means that the
amplitude and phase of reflection should be related to only one structural parameter (h_1
or w). To search for the condition of decoupled manipulation of reflection amplitude
and phase, we further calculate the coupling coefficients $\overline{M}_{A(\phi),h_1(w)}$ in the parameter
space (h, β) , as shown in the revised Fig. 1(b). From the figure, we clearly find the
existence of decoupled points (DPs) (*viz.*, $\overline{M}_{A,h_1} = 0$ and $\overline{M}_{\phi,w} = 0$) as well as quasi-
DPs (*viz.*, $\overline{M}_{A,h_1} \approx 0$ and $\overline{M}_{\phi,w} \approx 0$), where A can be regarded as being only related to
w , and ϕ only related to h_1 .

**Figure 1 | Decoupled modulation of reflection amplitude and phase.** (a) Schematic

diagram of holey metamaterials with an absorbing boundary at the back side, *viz.*, LAM.

3-D illustration and 2-D cross-section view of a unit cell are appended. (b) The coupling

coefficients $\overline{M}_{A(\phi),h_1(w)}$ versus h and β with DPs and quasi-DPs marked by the

crosses and arrows, respectively. (c) The reflection amplitude and phase responses to

parameters h_1 and w for a unit cell operating at quasi-DPs. (d-e) The simulated and

measured amplitude and phase versus w and h_1 , respectively, which reveals that the

reflection amplitude and phase are controlled by only one parameter, respectively.

For the revisions, please refer to **pages 5-6, lines 107-129** in the revised manuscript.

4) *Line 126 of page 7 has a discussion about the case where $\beta = 1$ and the fact that*
*it cannot be hit in reality because of the finite impedance contrast between air and*
*elastic solids. Isn't the $\beta = 1$ case where portions of the AMM structure is purely*
*air? The impedance contrast doesn't seem relevant.*

**Response:** We are sorry for not stating this issue clearly in the original version. Yes, the
condition $\beta = 1$ corresponds to the case where the channel wall is infinitely thin yet is
able to serve as a rigid boundary for providing total reflection to sound, and the LAM
structure, as indicated by the referee, is mathematically transformed into a trivial
structure of purely air. However, very thin channel walls are unavoidably flexible and
cannot be acoustically regarded as rigid unless they are made of solid with an infinitely
large acoustic impedance. This is physically unsound, since any practical solid must
have a finite rigidity and mass density, and we therefore think that the case of $\beta = 1$
could not be hit in reality and should be excluded.

In the revised manuscript, we have changed our expression on this point as
suggested by the referee. Please refer to “However, we cannot physically hit them due
to the fact that very thin channel walls are flexible and no longer provide a rigid
boundary (note that the rigidity of channel walls is the prerequisite condition of all our
derivations), and mathematically the whole LAM is transformed into a trivial structure
of purely air at \$\beta = 1\$.” on **pages 6-7, lines 130-133.**

5) *Aren't the patterns shown in Fig 1b Fabry-Perot types of resonances? Please address*
*in the revision.*

**Response:** Yes, the quasi-decoupling condition corresponds to the occurrence of Fabry-
Pérot resonances. In the revised manuscript, we add a comment on that. Please refer to
“Apparently, the quasi-decoupling condition corresponds to the occurrence of Fabry-
Pérot resonances.” on **page 7, lines 141-142.**

**Referee #3 (Remarks to the Author):**

*The paper “Hyperfine manipulation of sound via lossy acoustic metamaterials” by Zhu,*
*Hu, Fan, Yang, Liang, Zhu, and Cheng reports the manipulation of both the amplitude*
*and phase across the wavefront of an acoustic wave incident on a planar metamaterial*
*made of discrete sub-wavelength elements. The key feature reported is the introduction*
*of a loss (amplitude change) via controlled leaky emission from the backside of each*
*element. The authors successfully determine the requirements for independently setting*
*the complex amplitude and phase at each element, which leads to the demonstration of*
*holograms that can now encode both amplitude and phase. This is in principal an*
*interesting piece of work and an advance in the field of acoustics, as it suggests*
*improvements in the generation of sound fields. However, these improvements are*
*mainly shown in simulations and do not manifest themselves in the actual experiments.*
*Important information is missing and the claimed universal improvements are not*
*demonstrated. Therefore further work is needed and the authors are asked to address*
*the following points:*

**Response:** We thank the referee for the positive remarks and valuable advices. We have
made every effort to revise and improve the manuscript and added the important
missing information proposed by the referee.

1) Title: „hyperfine“ has a special meaning in physics. How does it relate to this work?
The authors probably mean high fidelity. However, the title should be changed.
Independent control of the static amplitude and phase across an acoustic wavefront is
the essence of this work and this should be reflected in the title.

**Response:** Thank you for pointing out those problems. We have made a careful check
throughout the manuscript and changed the term “hyperfine” into “fine” based on the
suggestion. In addition, we have changed the title into “Fine manipulation of sound via
lossy acoustic metamaterials with independently and arbitrarily distributed reflection”

amplitude and phase”.

2) *The approach the authors present is limited to reflection. The scalability, especially*
*miniaturization, is limited by two factors, (a) the fabrication method and (b) the*
*requirement of full absorption (or the disappearance) of transmitted wave components*
*at the backside. Considering these limitations the results are not “universal” and are*
*not as spectacular as the authors claim. The text should be changed accordingly.*

**Response:** Thank you for this valuable suggestion. In the revised manuscript, we have
discussed those limitations laid on our approach and changed the text accordingly as
suggested by the referee. For example, our approach can basically be extended into
projecting high-fidelity holograms in Fig. 2 as long as the number of unit cells is
sufficiently large for APM design. However, due to the size limitations in 3D-printing,
the pixel number on the hologram plane (119×119) in our experiment is much less than
the numerical investigations in Fig. 2 (359×359). We also discuss the case where the
back absorption is partial. To explore the device performance at partial backside
absorption, we first define a parameter of “image correlation” on the error analysis for
evaluating the quality of acoustic hologram. The “image correlation” measures the
degree of similarity between the numerical/experimental image and the target one,
where the mathematical definition can be referred to the Supplementary Note 5. Please
refer to

**“Note 5. Calculation of the correlation between two images.**

The correlation for evaluating the similarity between two images is calculated by

$$563 \text{Correlation} = \frac{\sum_m \sum_n (A_{mn} - \bar{A})(B_{mn} - \bar{B})}{\sqrt{\left(\sum_m \sum_n (A_{mn} - \bar{A})^2\right)\left(\sum_m \sum_n (B_{mn} - \bar{B})^2\right)}}, \quad (\text{S28})$$

where A and B are the data matrices of the two images, and \bar{A} and \bar{B} are the mean
values of the elements in the matrices A and B , respectively.” on pages 10-11, lines
136-140 in the supplementary materials. Based on the definition, a unitary correlation

denotes that the two images are identical, and the holographic image is perfect.

By utilizing the parameter of “image correlation”, we quantitatively investigate
performance of our approach when the back impedance is complex or not perfectly
absorbing. The correlations at different back impedances are shown in Fig. 5(i) in the
revised manuscript.

[Editorial Note: This image has been redacted to avoid copyright infringement.]

**Figure 5 | Experimental verification of single-plane 2-D acoustic hologram.** (a) The
pre-designed image of a tree. (b) Amplitude and phase profiles on the hologram plane
for projecting the tree image. (c) The photograph of the 3-D printed LAM sample. (d-
e) The simulated holographic image by the APM method and the experimentally
measured result. (f) The simulated holographic image by the PM method. (g) The
correlation between the resulting image and the pre-designed image at different
frequencies from 13kHz to 20kHz for APM method, and at 17kHz for PM method. (h)
The correlation between the resulting image and the pre-designed image when the image
plane locates at different distances. (i) The correlation between the resulting image and
the pre-designed image for different back impedances.

Note that the simulated holographic images in Figs. 5(d) and 5(f) are corresponding
 to the cases at the hollow triangle with back impedance $410\text{N}\cdot\text{S}/\text{m}^3$ (correlation=0.880)
 and at the red triangle (correlation=0.767) in Fig. 5(i), respectively. The results in Fig.
 5(i) show that we can still achieve very good holographic images (correlation>0.767)
 when the impedance at the back of the LAM structure is not perfectly absorbing. In this
 case, we can conduct simultaneous (but may not be independent) control of amplitude
 and phase, as unveiled in the added Fig. S5 of supplementary materials.

**Figure S5 | The calculated reflection amplitude and phase for different back**
 **impedances. (a) $Z=410\text{N}\cdot\text{S}/\text{m}^3$, (b) $Z=820+410i\text{N}\cdot\text{S}/\text{m}^3$, (c) $Z=1230+410i\text{N}\cdot\text{S}/\text{m}^3$.**

3) *The work mainly shows via simulations that the control of amplitude and phase*

607 *improves holograms. This is well known from optics. The paper does not appear to*
608 *demonstrate any (real) improvement in the experimental acoustic fields. A convincing*
609 *experimental demonstration is missing and should be provided by the authors so that*
*the importance of the work can be judged.*

**Response:** Thank you for pointing out this issue. In light of reviewer's important
suggestion, we have fabricated new samples of 119×119 unit cells and conducted
experiments on projecting a single-plane 2-D hologram of finer resolution in Fig. 5, and
multi-plane 3-D hologram in Fig. 6. Please refer to the added section on **pages 13-15**
in the revised manuscript, where we discuss the experimental verification of fine 2-D
hologram and multi-plane hologram in details.

**Experimental verification of single-plane 2-D hologram and multi-plane 3-D**
**hologram.** In this section, we choose a tree image [Fig. 5(a)] as our target object,
comprising 200×200 image pixels. Figure 5(b) presents the reflection amplitude and
phase profiles on the hologram plane for projecting the tree pattern in the far field. The
calculations of amplitude and phase profiles are based on Eq. (3). In the experiment, we
fabricated LAM samples via 3-D printing with precision of 0.1mm. The experiments
were carried out in an anechoic chamber to demonstrate the acoustic hologram
projection. We record both amplitude and phase information into the LAM sample,
where the sample size is $60 \times 60 \times 2 \text{ cm}^3$ with 119×119 unit cells, as shown by the photo
in Fig. 5(c). The size of image area is $60 \times 60 \text{ cm}^3$, with a distance 20cm away from the
surface of LAM. Other experimental details can be found in the Methods part. Due to
the size limitations in 3-D printing, the pixel number of the target image in our
experiment is less than the numerical investigations in Fig. 2.

We plot the simulated and measured intensity distributions on the image plane in
Figs. 5(d) and 5(e), respectively, showing a good agreement between numerical and
experimental results of fine 2-D hologram. Figure 5(f) shows the simulated result based
on the PM method for comparison. For a quantitative evaluation of the quality of
acoustic hologram, we introduce the parameter of "image correlation" which has been

commonly used for measuring the similarity between the numerical/experimental
image and the target one. The calculation of correlation can be referred to the
Supplementary Note 5. A higher value of correlation denotes a better similarity between
the generated holographic image and the target image, and only when the two images
are completely identical can a unitary correlation be achieved, which represents a
perfect hologram. Figure 5(g) shows the relation between image correlation and the
operation frequency. The results reveal that although our LAM is designed to work at
17kHz, it has a relatively broad operation bandwidth, thanks to the low dispersion of
the groove structure [33]. At 17kHz where the quasi-decoupled point (quasi-DP) locates,
the image correlation reaches a maximum of 0.880 in simulation, and the corresponding
measured data, albeit much lower than the simulated one due to the unavoidable
experimental error, still reaches 0.771 and is higher than the ideal value one can achieve
with PM method. We also note that the holographic image based on APM in a broad
frequency range (14kHz~20kHz, correlation>0.770) is better than the one of the PM
method (17kHz, correlation=0.767). The simulated holographic images at different
frequencies based on PM or APM are appended in the Supplementary Fig. 4. Notice
that our proposed method does not surpass standard resolution limits, which is in theory
the only limitation on its performance of sound manipulation. Hence the size of each
unit cell at the hologram plane is chosen as 1/4 wavelength, which is sufficiently small
for avoiding spatial alias and generating smooth phase and amplitude profiles. This
important feature, together with the independent control of magnitude and phase,
enables controlling acoustic waves with a higher fidelity control, especially when the
image plane is not far away from the sample. Figure 5(h) illustrates the comparison
between the image correlations as functions of the distances of holographic image
planes for APM and PM methods. Clearly, the hologram quality for APM is always
much better than that of PM regardless of the distance of image plane, although for both
cases the correlation slightly decreases with larger distances due to wave diffraction, as
shown in Fig. 5(h). Moreover, we emphasize simultaneous control of reflection
amplitude and phase can be achieved even when the back absorption is partial (see
Supplementary Fig. 5). In this case, we can also project holograms with relatively

higher correlations to the target image, as unveiled in Fig. 5(i).

At last, we demonstrated both numerically and experimentally the production of
precise distribution of acoustic energy in 3-D space. Here we choose to project the
acoustic hologram onto multiple planes instead of a single 2-D plane, as schematically
depicted at Fig. 6(a), where the holographic image is designed to be three hollow letters
“N”, “J”, and “U” at three different planes that are spacing 12cm, 16cm, 20cm away
from the hologram plane. The size of holographic regions at image planes 1, 2 and 3 is
$60\times 60\text{cm}^3$, and the bottom left corners of those holographic regions locate at (0cm,
30cm), (10cm, 0cm) and (30cm, 20cm) in the x - y plane. We record amplitude and phase
distributions [Fig. 6(b)] into the LAM sample of 119×119 unit cells [Fig. 6(c)]. By
comparing the amplitude field patterns in simulations and experiments, we
unambiguously observe a very good agreement. To be specific, the image correlations
to the perfect cases of letters “N”, “J”, “U” are 0.827(0.705), 0.867(0.771), 0.858(0.776)
for the results of simulations(experiments), respectively.

[Editorial Note: This image has been redacted to avoid copyright infringement.]

**Figure 5 | Experimental verification of single-plane 2-D acoustic hologram.** (a) The
pre-designed image of a tree. (b) Amplitude and phase profiles on the hologram plane
for projecting the tree image. (c) The photograph of the 3-D printed LAM sample. (d-
e) The simulated holographic image by the APM method and the experimentally
measured result. (f) The simulated holographic image by the PM method. (g) The
correlation between the resulting image and the pre-designed image at different
frequencies from 13kHz to 20kHz for APM method, and at 17kHz for PM method. (h)
The correlation between the resulting image and the pre-designed image when the image
plane locates at different distances. (i) The correlation between the resulting image and
the pre-designed image for different back impedances.

[Editorial Note: This image has been redacted to avoid copyright infringement.]

Figure 6 | Experimental verification of multi-plane 3-D acoustic hologram. (a) The
pre-designed image of a multi-plane acoustic hologram (Letters “N”, “J”, “U” at
different distances of 12cm, 16cm, 20cm). (b) Amplitude and phase profiles on the
hologram plane for projecting the “N”, “J”, “U” images at multiple planes. (c) The
photograph of the 3-D printed LAM sample. (d-e) The simulated holographic images

by the APM method and the corresponding experimentally measured results. The
correlations are marked in the figure. The correlation between the resulting image and
the predesigned image is appended below each sub-figure.

4) *Please, add scale bars or coordinate axes. This applies to almost all images and*
*plots.*

**Response:** Based on the referee's suggestion, we have added the scale bars on the
images and plots throughout the manuscript.

5) *How are the phase-only results (PM) obtained, against which the APM are compared?*
*Do you use an optimization procedure or simply keep the phase of the APM and reset*
*all amplitudes to 1? How does this compare to optimized PM of other published works?*
*This information must be provided.*

**Response:** Thank you for your questions and suggestions on this important issue. We
are sorry for not clarifying the details of the phase-only results shown in the original
version. In our work, we use an optimization procedure that is based on the Gerchberg-
Saxton (GS) algorithm commonly employed for producing pure-phase holograms in
other published works [see, e.g., Refs. 28-30]. In the revised manuscript, we have added
some discussions on this issue for clarification.

6) *It is not clear what "Freewheeling" means (abstract).*

**Response:** In the revised manuscript, we have changed the term "Freewheeling" into
"Fine". Please refer to **Page 2, line 24** in the revised manuscript.

7) *p.6, Equation 1: capital M is used for both coupling strengths and transfer matrices*
*in the SI. This is an unnecessary source of confusion and the nomenclature should be*
*changed.*

**Response:** In the revised supplementary materials, we have changed the capital “M”
 into capital “Q” to denote transfer matrices.

8) p.6, L.116: What does $(M_A) \overline{((M_\Phi)^{-1})} = 0$ mean? Is it $(M_A)^{-1} = (M_\Phi)^{-1} = 0$?

**Response:** Yes. To avoid possible misleading, we have changed “ $(M_A) \overline{((M_\Phi)^{-1})} = 0$ ”

into “ $\overline{M}_{A,h_1} = 0$ and $\overline{M}_{\phi,w} = 0$, respectively.” Please refer to the revision on **Page 6**,

**line 128** in the revised manuscript.

9) p.7, L.138: Do you mean Supplementary Note 3 or 4? Regarding Supp. Note 4, why

do you integrate w over $[0, 0.4]$ and h over $[0.2, 1.2]$? One would expect the ranges $[0,$

$\beta D]$ and $[0, \lambda/2]$, respectively.

**Response:** We thank the referee for pointing out this problem. Yes, for the integration,

the unit for the ranges is cm and the ranges are in fact $[0, \beta D]$ and $[0.1\lambda, 0.6\lambda]$,

respectively. We have fixed them in the revised version. Please refer to

“The coupling coefficients $\overline{M}_{A(\phi),h_1(w)}$ in the manuscript are calculated by

$$738 \quad \overline{M}_{A(\phi),h_1(w)} = \overline{M}_{A(\phi),h_1(w)}(\beta, h) / \max[\overline{M}_{A(\phi),h_1(w)}(\beta, h)], \quad (\text{S26})$$

where

$$740 \quad \begin{aligned} \overline{M}_{A,h_1}(\beta, h) &= \int_0^{\beta D} \int_{0.1\lambda}^{0.6\lambda} M_{A,h_1} dw dh_1 = \int_0^{\beta D} \int_{0.1\lambda}^{0.6\lambda} \frac{\partial A}{\partial h_1} dw dh_1, \\ \overline{M}_{A,w}(\beta, h) &= \int_0^{\beta D} \int_{0.1\lambda}^{0.6\lambda} M_{A,w} dw dh_1 = \int_0^{\beta D} \int_{0.1\lambda}^{0.6\lambda} \frac{\partial A}{\partial w} dw dh_1, \\ \overline{M}_{\phi,h_1}(\beta, h) &= \int_0^{\beta D} \int_{0.1\lambda}^{0.6\lambda} M_{\phi,h_1} dw dh_1 = \int_0^{\beta D} \int_{0.1\lambda}^{0.6\lambda} \frac{\partial \phi}{\partial h_1} dw dh_1, \\ \overline{M}_{\phi,w}(\beta, h) &= \int_0^{\beta D} \int_{0.1\lambda}^{0.6\lambda} M_{\phi,w} dw dh_1 = \int_0^{\beta D} \int_{0.1\lambda}^{0.6\lambda} \frac{\partial \phi}{\partial w} dw dh_1, \end{aligned} \quad (\text{S27})$$

on **Page 10, lines 130-133** in the supplementary materials.

10) On p.8 the authors write that “However, due to the lack of capability to modulate

both amplitude and phase, the current production of acoustic holograms ...cannot

*guarantee high-fidelity of images“. This does not seem to be correct as phase-only*
*holograms have been shown to generate extremely high-fidelity images?*

**Response:** Thank you for pointing out that. We have rewritten this paragraph. Please
refer to “However, due to the lack of capability to modulate both amplitude and phase,
the current production of acoustic hologram has to rely on phase-modulation (PM)
approaches combined with complex optimization process²⁴⁻³⁰.” on **Page 8, lines 165-**
**167** in the revised manuscript.

11) *p. 10, L.192ff: Please choose a number of unit cells that allows comparison to either*
*your experimental data or previously published hologram data. The images in Figure*
*2 are phenomenal but so is the element count of 359x359. The experimental data*
*presented in Figure 5 look mediocre compared to what has been achieved with pure*
*phase holograms in other works.*

**Response:** We thank the referee for the important suggestion. In light of the referee’s
suggestion, we have further increased the number of unit cells (albeit still much less
than the images shown in Fig. 2 due to the limitation on the size of our 3-D printing
machine) and added the experimental demonstration of projection of a 2-D image with
finer resolution as well as production of fine distribution of acoustic energy in 3-D space.
The renewed experimental results are shown in the updated Figs. 5 and 6 in the revised
manuscript, and some discussions have also been added on the comparison between our
proposed APM and PM methods. More details of this part can be referred to our reply
to question 3. Also, we have updated Fig. 2 by adding the results from optimized PM
and shown their comparisons to the results from APM method, which clearly verifies
the capability of our proposed scheme to generate very sophisticated acoustic
holograms with high fidelity that are challenging for PM.

In addition, we have provided a direct comparison to previously published hologram
data as suggested by the referee. Here we choose to use the proposed APM method to
produce the same holographic image as in Ref. 28 (Nature 537, 518–522 (2016)) and

show the results in Fig. R1. From Figs. R1(b) and (c), we can unambiguously see that
the quality of acoustic hologram generated by our APM method substantially
outperforms the result from PM employed in Ref. 28 (in the current stage an
experimental comparison is not technically feasible for us since the hologram in Ref.
28 was generated for ultrasound in water).

For the revision in the manuscript, please refer to “Here, we append the holographic
image simulated by PM optimization of Gerchberg-Saxton (GS) algorithm in Fig. 2(e).
Comparing Figs. 2(d) and 2(e), our method clearly outperforms the PM method,
providing a great flexibility in hologram reconstruction. The second target image is an
Einstein's photo with different gray values, where the amplitude at image pixels A_{0l} is
continuously changed between 0 and 1, as shown in Fig. 2(f). The holographic image
in Fig. 2(g) based on APM is consistent with the target image, while the holographic
image based on PM is very blurred, as shown in Fig. 2(h).” on **page 10, lines 207-214**
in the revised manuscript.

**[Editorial Note: This image has been redacted to avoid copyright infringement.]**

**Figure 2 | High-fidelity acoustic hologram.** (a) Schematic diagram of hologram
reconstruction. (b) Schematic diagram of how LAM projects high-quality acoustic
hologram in simulation and experiment. (c) The target image of a school badge with
complex amplitude distributions. (d) The simulated holographic image by the APM
method. (e) The simulated holographic image by the PM method. (f-h) Another case of
projecting a more complicated acoustic hologram with the target image an Einstein's
photo. The simulation is conducted by effective parameters.

**[Editorial Note: This image has been redacted to avoid copyright infringement.]**

Fig. R1. (a) Target image in Ref. 28. (b) Numerical simulation with APM method in
this work. (c) Numerical simulation with PM method in Ref. 28 (Nature 537, 518–522
(2016)).

12) *p.12, L.245: Reference to equation 3 not 5.*

**Response:** Thank you for pointing it out. In the revised manuscript, we have fixed them.

13) *p.13, L.265: The Penrose pattern is shown in Figure S3.*

**Response:** Thank you for pointing it out. In the revised manuscript, the Penrose pattern
is replaced by new experiments as shown in Figs. 5 and 6.

14) *At various locations throughout the manuscript and in the conclusions the authors*
*speak of “modulating both amplitude and phase of acoustic wave in a precise,*
*continuous and decoupled manner”. This is somewhat misleading as continuous*
*modulation suggests a temporal or dynamic control. The authors should clarify this by*
*stating clearly in the text that they only consider fixed or static acoustic holograms.*

**Response:** Thank you for pointing it out. In the revised manuscript, we have fixed it
into “modulating both amplitude and phase of acoustic waves in a static, precise, and
decoupled manner.” on **Page 16, lines 331-332** in the revised manuscript.

Reviewers' comments:

Reviewer #1 (Remarks to the Author):

The authors have submitted an improved manuscript, now with additional experiments, explanations and data analysis. In the opinion of the reviewer, all the raised points have been addressed and the reviewer is happy to support its publication.

Reviewer #2 (Remarks to the Author):

The authors have provided satisfactory revisions to this manuscript to merit publication. I recommend acceptance of the current submission.

Reviewer #3 (Remarks to the Author):

The authors have implemented many of my suggestions. However, a few comments remain, including my main criticism that Fig. 1 is misleading, as it suggests something the authors have not managed to demonstrate, and that the figure should therefore be replaced with one that corresponds to the level of complexity that was experimentally demonstrated. I have the following comments for the authors to consider:

- 1) Please chose images for Fig. 1 that are of comparable complexity to the experimental work demonstrated, e.g. the university logo.
- 2) Can you explain why the quality difference between the APM and PM reconstructions are so much higher in Figure 1 than in Figure 5?
- 3) There are some unit errors in the text. In the main text areas should be in cm^2 (lines 274 and 320) and in the SI please correct $\text{N}^*\text{s}/\text{m}^3$ (small letter s)
- 4) In line 282 the authors write [...] we introduce the parameter of "image correlation" which has been commonly used for measuring the similarity between the numerical/experimental image and the target one." Please cite an appropriate reference for this claim.

Referee #1 (Remarks to the Author):

The authors have submitted an improved manuscript, now with additional experiments, explanations and data analysis. In the opinion of the reviewer, all the raised points have been addressed and the reviewer is happy to support its publication.

Response: We sincerely thank the referee for recommending our work to be published in **Nature Communications**.

=====

Referee #2 (Remarks to the Author):

The authors have provided satisfactory revisions to this manuscript to merit publication. I recommend acceptance of the current submission.

Response: We sincerely thank the referee for recommending our work to be published in **Nature Communications**.

=====

Referee #3 (Remarks to the Author):

The authors have implemented many of my suggestions. However, a few comments remain, including my main criticism that Fig. 1 is misleading, as it suggests something the authors have not managed to demonstrate, and that the figure should therefore be replaced with one that corresponds to the level of complexity that was experimentally demonstrated. I have the following comments for the authors to consider:

1) Please chose images for Fig. 1 that are of comparable complexity to the experimental work demonstrated, e.g. the university logo.

Response: Thank you for your important suggestion. We have chosen the figure of the university logo by following the referee's suggestion. The much more complicated image of Einstein's photo is moved to Supplementary Figure 4.

[Editorial Note: This image has been redacted to avoid copyright infringement.]

Figure 2 | High-fidelity acoustic hologram. (a) Schematic diagram of hologram reconstruction. (b) Schematic diagram of how LAM projects high-quality acoustic hologram in simulation and experiment. (c) The target image of a school logo with complex amplitude distributions. (Scale bar, 20cm) (d) The simulated holographic image via the APM method. (e) The simulated holographic image via the PM method.

[Editorial Note: This image has been redacted to avoid copyright infringement.]

Supplementary Figure 4 | Simulations for a complicated image of Einstein's photo. (a) The target image with a complex amplitude distribution. (Scale bar, 20cm) (b) The generated holographic image via the APM method. (c) The generated holographic image via the PM method.

2) Can you explain why the quality difference between the APM and PM reconstructions are so much higher in Figure 1 than in Figure 5?

Response: Thank you for this enlightening question. As we know, the complete information of sound field includes amplitude and phase. In light of time-reversal symmetry, it is hence necessary to modulate both the reflection amplitude and phase for achieving an exact hologram reconstruction of a complex image, while the pure-phase scheme is innately unable to perfectly reconstruct the target image due to the lack of amplitude information on the hologram plane. For relatively simple images such as the pattern in Fig. 5(a), the PM method leads to less obvious errors as shown in Fig. 5(f),

due to the amplitude distribution via the APM method on the hologram plane is relatively uniform as shown in Fig. 5(b), albeit the improvement by APM in Fig. 5(d) is still evident by both the naked-eye visual effect and the quantitative evaluation of “image correlation”. However, the image error caused by limiting a uniform amplitude distribution on the hologram plane becomes quite prominent when the target image is complicated and comprises a large number of pixels with uneven amplitude levels. That is the crux responsible for the much higher quality difference between the APM and PM reconstructions as shown in Figs. 2 and 5, which also proves the unique advantage of our proposed approach.

Following the suggestion of the referee, we have added a brief clarification on the higher quality difference between the APM and PM reconstructions in Figs. 2 and 5. Please refer to “Comparing Figs. 2(d) and 2(e), our APM method clearly outperforms the PM method, since in light of time-reversal symmetry it is necessary to modulate both the reflection amplitude and phase for achieving an exact hologram reconstruction of a complex image. We also note that in Fig. 2(e), the image error caused by limiting a uniform amplitude distribution on the hologram plane becomes quite prominent when the target image is complicated and comprises a large number of pixels with uneven amplitude levels. The result demonstrates the effectiveness and flexibility of our method in complicated hologram reconstruction. Simulations for a more complicated hologram (*e.g.*, Einstein’s photo) are provided in Supplementary Fig. 4 to further reveal the advantage of APM method .” on Page 10, lines 199-208 and “For relatively simple images such as the pattern in Fig. 5(a), the PM method leads to less obvious errors as shown in Fig. 5(f), due to the amplitude distribution via the APM method on the hologram plane is relatively uniform as shown in Fig. 5(b), albeit the improvement by APM in Fig. 5(d) is still evident by both the naked-eye visual effect and the quantitative evaluation of “image correlation”.” on Page 13, lines 281-286.

3) *There are some unit errors in the text. In the main text areas should be in cm² (lines 274 and 320) and in the SI please correct N*s/m³ (small letter s)*

Response: Thanks for pointing out the unit errors. We have fixed them.

4) *In line 282 the authors write [...]we introduce the parameter of “image correlation” which has been commonly used for measuring the similarity between the numerical/experimental image and the target one.” Please cite an appropriate reference for this claim.*

Response: We have added the reference “34. Lewis, J. Fast template matching. *Vision interface* **95**, 15-19 (1995)”, which is cited at “For a quantitative evaluation of the quality of acoustic hologram, we introduce the parameter of “image correlation” which has been commonly used for measuring the similarity between the numerical/experimental image and the target one³⁴.” in lines 275-278 on page 13.

REVIEWERS' COMMENTS:

Reviewer #3 (Remarks to the Author):

The authors have responded to my comments and made appropriate changes to their manuscript.
Thank you.